Primate tarsal bones from Egerkingen, Switzerland, attributable to the middle Eocene adapiform Caenopithecus lemuroides

Seiffert Erik R. 1 erik.seiffert@stonybrook.edu
Costeur Loïc 2
Boyer Doug M. 3 douglasmb@gmail.com
1 Department of Anatomical Sciences, Stony Brook University , Stony Brook, NY , USA
2 Department of Geosciences, Naturhistorisches Museum Basel , Basel , Switzerland
3 Department of Evolutionary Anthropology, Duke University , Durham, NC , USA
Kraatz Brian
Electronic publication date: 2015 Jun 23
Publication date: 2015
Volume: 3
Electronic Location ID: e1036
Received 2015 Mar 25; Accepted 2015 May 29
Copyright: © 2015 Seiffert et al.
Copyright year: 2015
Copyright holder: Seiffert et al.
License: This is an open access article distributed under the terms of the Creative Commons Attribution License, which permits unrestricted use, distribution, reproduction and adaptation in any medium and for any purpose provided that it is properly attributed. For attribution, the original author(s), title, publication source (PeerJ) and either DOI or URL of the article must be cited.
License URL: https://creativecommons.org/licenses/by/4.0/

Keywords: Adapidae, Afradapis, Talus, Strepsirrhini, Calcaneus, Astragalus, Europe, Africa, Locomotion, Phylogeny

Funding: U.S. National Science Foundation BCS 1317525 BCS 1125507 U.S. National Science Foundation BCS 1304045 This work was funded by U.S. National Science Foundation grant BCS 1317525 (formerly BCS 1125507) to DMB and ERS. Inclusion of dental material and analysis of that material was supported by U.S. National Science Foundation grant BCS 1304045 to DMB and Elizabeth M. St. Clair. The funders had no role in study design, data collection and analysis, decision to publish, or preparation of the manuscript.

==============================
The middle Eocene species Caenopithecus lemuroides, known solely from the Egerkingen fissure fillings in Switzerland, was the first Paleogene fossil primate to be correctly identified as such (by Ludwig Rütimeyer in 1862), but has long been represented only by fragmentary mandibular and maxillary remains. More recent discoveries of adapiform fossils in other parts of the world have revealed Caenopithecus to be a biogeographic enigma, as it is potentially more closely related to Eocene adapiforms from Africa, Asia, and North America than it is to any known European forms. More anatomical evidence is needed, however, to provide robust tests of such phylogenetic hypotheses. Here we describe and analyze the first postcranial remains that can be attributed to C. lemuroides—an astragalus and three calcanei held in the collections of the Naturhistorisches Museum Basel that were likely recovered from Egerkingen over a century ago. Qualitative and multivariate morphometric analyses of these elements suggest that C. lemuroides was even more loris-like than European adapines such as Adapis and Leptadapis, and was not simply an adapine with an aberrant dentition. The astragalus of Caenopithecus is similar to that of younger Afradapis from the late Eocene of Egypt, and parsimony and Bayesian phylogenetic analyses that include the new tarsal data strongly support the placement of Afradapis and Caenopithecus as sister taxa to the exclusion of all other known adapiforms, thus implying that dispersal between Europe and Africa occurred during the middle Eocene. The new tarsal evidence, combined with previously known craniodental fossils, allows us to reconstruct C. lemuroides as having been an arboreal and highly folivorous 1.5–2.5 kg primate that likely moved slowly and deliberately with little or no capacity for acrobatic leaping, presumably maintaining consistent powerful grasps on branches in both above-branch and inverted postures.

Introduction

Caenopithecus is a phylogenetically and biogeographically enigmatic adapiform primate whose fossil record is restricted to middle Eocene (Lutetian, ∼43 Ma, MP13b) fissure fillings in the commune of Egerkingen, Canton Solothurn, northern Switzerland (Godinot, 1998; Rütimeyer, 1862; Stehlin, 1916). Upper molars of the type and only species, Caenopithecus lemuroides, were described by Rütimeyer (1862), and later Stehlin (1916) described parts of the lower dentition, mandible, additional parts of the upper dentition, and the orbital region. These limited remains show that C. lemuroides was a relatively large adapiform—having second lower molars that are about the same length as those of the extant lemurids Eulemur fulvus and Prolemur simus (Kay et al., 2004)—and had a fused mandibular symphysis, large canine teeth, very small P2/2, simple P3−4/3−4, quadrate upper molars with distinct hypocones and mesostyles, and narrow lower molars with elongate crests and well-developed metastylids. The maxillary remains of C. lemuroides show that the species had a postorbital bar, but no postorbital closure. Analysis of lower molar shearing crests suggests that C. lemuroides was a dedicated folivore (Kay et al., 2004).

Other remains of Caenopithecus have been reported since the publication of Stehlin (1916), but are not universally accepted as belonging to that genus—Franzen (1994) identified partial skeletons from the Geiseltal and Messel deposits as belonging to a new species of Caenopithecus (i.e., Caenopithecus “neglectus”), but these are now the type specimens of Godinotia neglecta (Franzen, 2000) and Darwinius masillae (Franzen et al., 2009), and Godinot (1988) has expressed doubt about the presence of Caenopithecus at Bouxwiller, France, that was reported by Jaeger (1971). Stehlin also attributed an astragalus from Egerkingen (NMB Eh 741) to C. lemuroides, but this element is probably that of a non-primate hyaenodontidan ‘creodont’ (see also Decker & Szalay, 1974).

In an unpublished doctoral dissertation, Dagosto (1986) attributed an astragalus and three calcanei from the original Egerkingen collections held at the Naturhistorisches Museum Basel (NMB) to C. lemuroides and briefly described those elements.1 Here we expand on Dagosto’s earlier work by (1) providing quantitative justification for attribution of these tarsals to C. lemuroides, (2) describing and comparing the Egerkingen tarsals in greater detail, notably making comparisons with several specimens that have been discovered since Dagosto’s work was completed in 1986; (3) analyzing the more complete specimens using multivariate and other morphometric techniques; and (4) including C. lemuroides in phylogenetic analyses alongside numerous other living and extinct primates. The tarsals of C. lemuroides provide important new insights into the locomotor adaptations and relationships of this mysterious primate.

History of study

Rütimeyer (1862) identified Caenopithecus as a primate (and in so doing was the first to correctly identify a fossil primate of Paleogene age (Stehlin, 1916)); with little comparative material available for study, he (and later Forsyth Major, 1872) was impressed by upper molar features that Caenopithecus shared with extant howler monkeys (Alouatta). The identification of Caenopithecus as a primate was subsequently questioned by various authorities (e.g., Delfortrie, 1873), but Gervais (1873) considered Caenopithecus to be a synonym of Adapis (see also Flower, 1876; Forbes, 1894; Schlosser, 1887), and Gaudry (1878) argued that Caenopithecus was aligned with lemurs, drawing attention to dental features that the genus shared with extant Hapalemur and Lemur. Stehlin (1916) noted additional features of the lower dentition and mandible that Caenopithecus shared with the few additional European adapiforms that were known by the time of his study. With the benefit of an expanding North American fossil record, Stehlin also pointed out dental similarities to Shoshonius and Washakius, which are now considered to be distantly related washakiin omomyiforms. Abel (1931) and Le Gros Clark (1959) believed Caenopithecus to be a tarsioid,2 and Gregory (1920) later suggested that the genus might be related to indrioid lemurs, but these are minority views; most debates subsequent to the publications of Stehlin (1912) and Stehlin (1916) have focused on whether Caenopithecus is more closely related to adapines such as Adapis and Leptadapis (e.g., Godinot, 1998; Rose, Godinot & Bown, 1994; Russell, Louis & Savage, 1967; Simons, 1972; Simpson, 1940; Szalay & Delson, 1979) or to some “cercamoniine” adapiform such as Europolemur or Protoadapis (Franzen, 1994; Gingerich, 1977; Remane, 1956; Simons, 1962; Weigelt, 1933). In her unpublished doctoral dissertation, Dagosto (1986) briefly described and figured the tarsals analyzed here, attributed them to C. lemuroides, and concluded that they “support the hypothesis of a close relationship between Adapis, Leptadapis, and Caenopithecus” (p. 198) and that “Adapis, Leptadapis, and Caenopithecus…have abandoned the primitive euprimate emphasis on leaping and have evolved locomotor modes which stress quadrupedal locomotion and climbing” (pp. 201–202).

The study of Caenopithecus has been further complicated in recent years by the discovery of several geographically far-flung taxa, such as Aframonius, in the late Eocene of Africa (Simons, Rasmussen & Gingerich, 1995), and the poorly known Adapoides, in the middle Eocene of Asia (Beard et al., 1994), that have been identified as close relatives of Caenopithecus and placed in Caenopithecinae (a subfamily of the otherwise strictly European Adapidae; Godinot (1998)). In proposing this group, Godinot (1998) also included the problematic middle Eocene genus Mahgarita from North America. Membership has since expanded to include late Eocene Afradapis from Egypt (Boyer, Seiffert & Simons, 2010; Seiffert et al., 2009), and possibly Mescalerolemur (from the middle Eocene of Texas; Kirk & Williams, 2011) and Darwinius (from the middle Eocene of Europe; Franzen et al., 2009; see phylogenetic analysis of Seiffert et al., 2009). Mahgarita and Mescalerolemur are the only non-notharctine adapiforms known from North America, and, like Caenopithecus, their geographic origins remain mysterious (Kirk & Williams, 2011). Fleagle (2013) placed all of these taxa, including middle Eocene European Europolemur and Godinotia, into a new family, Caenopithecidae. Here we use the nomen Caenopithecinae and use it to refer to Caenopithecus and its demonstrable near relatives (Afradapis and Aframonius), because the phylogenetic analyses that we present here call into question the monophyly of an assemblage that includes these taxa as well as Mahgarita and Mescalerolemur (see also Kirk & Williams, 2011). We refer to the clade that includes Adapis and Leptadapis (the monophyly of which is consistently supported by our phylogenetic analyses) as Adapinae.

The higher-level affinities of caenopithecines and Mahgarita continues to be debated; authorities have identified some of these species as stem anthropoids or stem haplorhines (Franzen et al., 2009; Rasmussen, 1990; Rasmussen, 1994; Simons, Rasmussen & Gingerich, 1995) while others have considered some or all of these species to be most parsimoniously interpreted as stem strepsirrhines (Dagosto & Gebo, 1994; Godinot, 1998; Kay, Ross & Williams, 1997; Kirk & Williams, 2011; Maiolino et al., 2012; Marivaux et al., 2013; Ni et al., 2013; Rose et al., 2009; Ross, Williams & Kay, 1998; Seiffert et al., 2009; Seiffert et al., 2005). Until recently, the only evidence that could be brought to bear on this debate was the dental anatomy of these taxa, and the poorly preserved crania of Mahgarita (see for instance the different interpretations of Rasmussen, 1990, and Ross, 1994), but the recent discovery of an astragalus of Afradapis (Boyer, Seiffert & Simons, 2010) showed that taxon to be remarkably strepsirrhine-like in its tarsal morphology, contrasting strongly with that which would be expected along the haplorhine or anthropoid stem lineages (see for instance Boyer & Seiffert, 2013). The astragalus of Afradapis shows several similarities to the astragalus from the Egerkingen collections that we describe here, bolstering the evidence for its attribution to Caenopithecus, which has been placed as the sister taxon of Afradapis in multiple phylogenetic analyses (Boyer & Seiffert, 2013; Boyer, Seiffert & Simons, 2010; Kirk & Williams, 2011; Ni et al., 2013; Seiffert et al., 2009). We further test all of these phylogenetic hypotheses by adding the character data from the Egerkingen tarsals to the morphological character matrix that has most recently been employed by Boyer et al. (2015b), which we analyze using both parsimony and Bayesian approaches.

Materials and Methods

Attribution

The tarsals described here derive from the Egerkingen fissure fills, but provenance is only documented for one specimen, NMB Eh 719, which was recovered from the γ (= “Gamma”) fissure. In her dissertation, Dagosto (1986) stated that all of the specimens are from the γ fissure, suggesting that locality information for the specimens was available in 1986 but has since been lost. On the basis of articular compatibility of the astragalus (NMB En.270) with the calcaneus NMB Eh 719, as well as the similar size, color, preservation, and peculiar morphology of all three calcanei (i.e., NMB Eh 719, En.268, and En.269), we consider it highly probable that all of the tarsals described here belong to the same species, and so are likely to derive from the same fissure that Eh 719 was recovered from (i.e., fissure γ, where the two relatively large adapiforms C. lemuroides and Leptadapis priscus occur; note that Dagosto (1986), was unaware that remains of L. priscus have also been recovered from fissure γ, and so thought that the much smaller species Microadapis sciureus was the only other option for attribution). We employ measures of relative abundance (by comparing the number of craniodental specimens attributed to each large adapiform species known from Egerkingen γ) and regressions of dental and tarsal variables (i.e., astragalar trochlear width and calcaneo-cuboid joint size to M2 size across a sample of living primates) to determine the most probable species attribution. Original scans and digital models of all specimens are available on MorphoSource (www.morphosource.org); a DOI for each specimen is provided in Table 1.

Table 1 DOIs and MorphoSource ID numbers for scans and digital models of the Egerkingen tarsals.

DOI	Morphosource ID	File name	File type	Specimen number	Element	
10.17602/M2/M5397	M5963-5397	NMB-En-270_M5963-5397.zip	Zipped tiff stack	NMB En.270	Left astragalus	
10.17602/M2/M5398	M5963-5398	NMB-En-270_M5963-5398.ply	Smoothed mesh file	NMB En.270	Left astragalus	
10.17602/M2/M5399	M5964-5399	NMB-Eh-719_M5964-5399.zip	Zipped tiff stack	NMB Eh 719	Right calcaneus	
10.17602/M2/M5400	M5964-5400	NMB-Eh-719_M5964-5400.ply	Smoothed mesh file	NMB Eh 719	Right calcaneus	
10.17602/M2/M5401	M5965-5401	NMB-En-268_M5965-5401.zip	Zipped tiff stack	NMB En.268	Right calcaneus	
10.17602/M2/M5402	M5965-5402	NMB-En-268_M5965-5402.ply	Smoothed mesh file	NMB En.268	Right calcaneus	
10.17602/M2/M5403	M5966-5403	NMB-En-269_M5966-5403.zip	Zipped tiff stack	NMB En.269	Left calcaneus	
10.17602/M2/M5404	M5966-5404	NMB-En-269_M5966-5404.ply	Smoothed mesh file	NMB En.269	Left calcaneus	

We gathered comparative data on tarsal facet size and M2 size from extant and fossil primates (Appendix S1 and S2) in order to determine whether the Egerkingen tarsal bones predict tooth sizes more similar to those of C. lemuroides or L. priscus. To this end, we regressed natural log-transformed lower second molar (M2) area (= maximum mesiodistal length multiplied by maximum buccolingual breadth) on natural log-transformed astragalar trochlear width (shown to be highly correlated with body mass in Dagosto & Terranova (1992: their measure “A4”)) and calcaneo-cuboid facet area (= cuboid facet length times cuboid facet width, another measure shown to be highly correlated with body mass by Dagosto & Terranova (1992: their “Index 6”) and the best calcaneal predictor of body mass found by Yapuncich, Gladman & Boyer (2015)). Though M2 area may not be as robust a body mass proxy as M1 area (Gingerich, Smith & Rosenberg, 1982), it is a measure that is known in C. lemuroides and L. priscus, and is arguably more removed from functional/evolutionary pressures on tooth size and morphology that affect the adjacent molars. We also estimate the body mass of the Egerkingen primate using the equations derived from regressions of tarsal facet areas on body mass that were recently published by Yapuncich, Gladman & Boyer (2015).

Our null hypothesis for these analyses was that the Egerkingen tarsals belong to C. lemuroides, on the basis of the expression of morphology that would be consistent with that species’ demonstrated phylogenetic proximity to A. longicristatus, but not clearly consistent with attribution to a primitive adapine such as L. priscus, whose close relatives (including the alleged congener of L. priscus, Leptadapis magnus) differ in clear and quantifiable ways. If true, the tarsal dimensions (i.e., those of the astragalus NMB En.270 and the best-preserved calcaneus from Egerkingen, NMB Eh 719) should predict an M2 size matching that of teeth identified as belonging to C. lemuroides. Our extant sample includes 33 species means, in which all individuals used in computing those means included molar, astragalar, and calcaneal measurements (except for Galagoides demidoff, for which the molar data came from different specimens than those that provided the tarsal data). The sample also includes 30 fossil taxa, of which nine specimens have associated molar and tarsal data (Appendix S2). We also computed 95% prediction intervals as the limiting envelope within which the actual molar area must fall to be considered plausibly attributable to the postcranial specimen generating the estimate. Regressions were run in PAST.exe (Hammer, Harper & Ryan, 2001). 95% confidence limits on the prediction interval of tooth size from postcranial element dimensions were generated (using equation 17.29 of Zar, 1984). Note that prediction intervals that take into account the Prediction Standard Error (PSE) of the data (or the ‘scatter’) around the regression line are more appropriate than a confidence interval based on standard error in the regression parameters (e.g., Chatterjee & Simonoff, 2013). Prediction intervals intuitively must be broad enough to incorporate most of the data points used to construct the regression, whereas confidence intervals based on error in regression parameters typically exclude many more data points used to create the regression, indicating that these limits are not appropriate indicators of whether a given set of (for instance) postcranial and molar dimensions are closely associated.

In plotting adapiform M2 area and tarsal measurements among those of other taxa (Fig. 1), we observed that larger species (i.e., those at or above “Kay’s Threshold” of 500 g) tend to have higher residuals. Therefore, instead of expecting the “owner” of NMB En.270 and NMB Eh 719 to have a small residual between its actual value and predicted value, we reasoned that it would more likely to exhibit a residual value close to the average residual value shown by other large adapiforms in the sample. To compare residuals among adapiforms, we first ran a regression that excluded large adapiforms (i.e., ten taxa were excluded from the trochlear width regression, and eight were excluded from the calcaneo-cuboid facet regression). We re-computed adapiform residuals relative to this new line and then compared the values of the M2 area residuals generated by the tarsal specimens under investigation to these populations using one sample t-tests, where the residual values of the tarsal dimensions were treated as the test values.

Figure 1 Regression of tooth size on tarsal facet size.

Above, regression of the natural log of lower second molar area (length times width) on the natural log of calcaneo-cuboid facet area (length times width), with data from 51 living and extinct primates and seven non-primate euarchontans. Solid circles, extant non-primate euarchontans; open squares, plesiadapiforms; solid squares, tarsiers and extinct omomyiforms; open triangles, extinct adapiforms; open circles, extant lemuriforms; “x,” extant galagids; open diamonds, extant lorises; “+,” natural log of calcaneo-cuboid facet area of NMB Eh 719 plotted relative to the natural log of second lower molar size of Leptadapis priscus; “+” enclosed in a diamond, natural log of the calcaneo-cuboid facet area of NMB Eh 719 plotted relative to the natural log of second lower molar size of Caenopithecus lemuroides. See figure for regression equation and r2. Below, regression of the natural log of lower second molar area (length times width) on the natural log of astragalar trochlear width, with data from 63 living and extinct primates and seven non-primate euarchontans. Symbols as above; “+,” natural log of the trochlear width of NMB En.270 plotted relative to the natural log of second lower molar size of Leptadapis priscus; “+” enclosed in a diamond, natural log of trochlear width of NMB En.270 plotted relative to the natural log of second lower molar size of Caenopithecus lemuroides. See figure for regression equation and r2. Note that the extant pen-tailed tree shrew Ptilocercus shows a tooth-tarsal scaling relationship that differs from that of living and extinct primates; for this reason it was excluded from the regression equation.

Finally, to address the possibility that the astragalus NMB En.270 and the calcaneus NMB Eh 719 could have come from two different species rather than a single species, we noted that for dentally-associated elements, molar residuals from the calcaneus and astragalus were strongly correlated. To quantify this correlation we converted residuals from the original regressions into z-scores. For each taxon occurring in both regressions (n = 57), we computed the absolute value of the differences between residuals generated from the trochlear width and calcaneo-cuboid joint area regressions. We found these differences to be small in general, and hypothesized that if both tarsals came from a single species, the difference in M2 residuals should be within the 95% confidence interval of the differences exhibited by the extant sample. If the tarsals came from two different species, there is no way to constrain how much difference should exist between the two sets of residuals when the bones are treated as if they came from a single species. Thus this test can potentially refute the hypothesis that both bones came from a single species (if the difference is outside the observed range), but not that they came from two species.

Multivariate morphometric analysis of primate astragali

In order to quantitatively assess the phenetic affinities of the Egerkingen tarsals, we undertook a principal components analysis of a set of linear and angular measurements taken on digital models of 25 primate and 27 non-primate astragali. The dataset was first developed by Boyer (2009), and later augmented by Boyer, Seiffert & Simons (2010) and Chester et al. (2015). The astragalus described here (NMB En.270) lacks most of the head, and is abraded in such a way that not all measurements in the dataset could be taken. From an original set of 18 linear measurements, we were able to take 11 (specifically 2, 4–8, 10–14 from Boyer, Seiffert & Simons, 2010, some of which (4, “Fibular facet maximum dorsoplantar height”; 5, “Fibular facet proximodistal length”; 7, “Lateral tibial facet maximum mediolateral width”; and 12, “Flexor fibularis groove mediolateral width”) were estimated due to abrasion along the lateral margin of the lateral tibial facet (for 4, 5, and 7), and along the plantar surface of the medial tubercle buttressing the groove for flexor fibularis (for 12). Of six angular measurements, we were able to take three (20–22) (Appendix S3). All linear measurements were converted to shape ratios by dividing each measurement by a geometric mean (based on 10 measurements—4–8, 10–13, 15) and then log transforming those ratios. Angular measurements were converted to radians. A principal components analysis of these data was undertaken using the program PAST. See Appendix S3 for the complete dataset.

Quantification of flexor fibularis groove depth

We used digital models of the astragali of 52 crown strepsirrhine individuals (20 genera, including three subfossil lemuriform genera) and seven adapiform genera to quantify the depth of the flexor fibularis groove along the most anterior extent of its plantar exposure. Astragali were oriented with the plantar surface facing upward and in posterior view, such that the point marking the base of the trough of the flexor fibularis groove was aligned with the point marking the plantar apex of the navicular facet or sustentacular facet (whichever was visible in that view). This cross-sectional view of the flexor fibularis groove was therefore oriented roughly perpendicular to the anteroposteriorly oriented line of action of the flexor fibularis tendon along the plantar surface of the astragalus. In this view, the two peaks formed by (1) the medial tubercle buttressing the groove and (2) the most plantar projection of the anteromedial aspect of the ectal facet provided landmarks for two measurements that were taken simultaneously using the “Measure” tool in the program Geomagic—the linear width of the flexor fibularis groove, and (by clicking “Projection” in Geomagic) the length of the contour between the two points that were used to calculate linear width (which, in Geomagic, is not dependent on orientation but rather is the shortest distance between those two points along the contour). Surfaces with rendering artifacts, such as artificially roughened or “spikey” areas, were smoothed in Geomagic to ensure accuracy of the contour measurement. We used a simple ratio of the contour measurement to the linear width measurement to describe the depth of the groove; species with a ratio of 1 show no concavity of the groove, while increasing positive departures from that value reflect increasing concavity (and therefore tall walls constraining the passage of the tendon). In species with very little concavity of the groove, the two landmarks could be difficult to place (because there were no obvious “peaks”), but this is not of great concern to us because wherever the two points were placed in such species, the ultimate values for the ratio approached equality; our interest in taking this measurement was to detect marked departures from equality, and to identify taxa that had markedly concave flexor fibularis grooves along the plantar surface of the astragalus.

Peroneal tubercle position in early fossil primates

We took three measurements along the proximodistal long axis of the calcaneus in order to quantify peroneal tubercle position and size across a sample of early fossil primates (1: length of the proximal segment; 2: distance from the proximal-most aspect of the calcaneal tuber to the distal-most projection of the peroneal tubercle; and 3: distance from the proximal-most aspect of the calcaneal tuber to the midpoint of the peroneal tubercle). The sample includes a total of 100 individuals, composed of 51 adapiform specimens (Adapis (n = 6), Asiadapis (n = 2), Caenopithecus (n = 3), Cantius (n = 16), Leptadapis (n = 8), Marcgodinotius (n = 5), Notharctus (n = 7), Smilodectes (n = 4)), 21 omomyiform specimens (Arapahovius (n = 3), Hemiacodon (n = 1), Omomys (n = 6), Ourayia (n = 1), Shoshonius (n = 1), Teilhardina (belgica, n = 8), Tetonius (n = 1)), nine stem anthropoid specimens (Eosimias (n = 6), Parapithecidae (n = 5), Proteopithecus (n = 1)), six plesiadapiform specimens (Carpolestes (n = 1), Ignacius (n = 1), Nannodectes (n = 1), Plesiadapis (n = 3)), three dermopteran specimens (Cynocephalus (n = 2), Galeopterus (n = 1)), and ten scandentian specimens (Ptilocercus (n = 3), Tupaia (n = 7)).

Automated geometric analysis of primate calcanei

In order to compare overall shape of the best-preserved calcaneus from Egerkingen (NMB Eh 719) with that of other living and extinct primates, we used an automated morphometric procedure that requires no researcher supervision (i.e., no measurements, landmarks or anatomical axes need be supplied for bones included in the comparison) (Boyer et al., 2015a). We chose to take this approach in order to minimize the degree to which characterizations of shape affinities are dependent on measurements selected, or researcher observer error or bias. We would opt for this approach with the astragalus as well, but the method cannot be easily implemented for analysis of fragmentary bones at this time. In order to create 3D digital models of calcanei, 159 specimens representing 46 primate genera and 6 non-primate euarchontan genera were Micro-CT or laser scanned and processed in Avizo and Geomagic to create shell-like (i.e., without internal structure) mesh files representing only the external surface of each bone. All surface files are published on www.morphosource.org and can be directly downloaded, though the cleaned, shell-like versions are not necessarily represented (but are available on request). These 3D digital models were then analyzed using the fully automated 3D geometric morphometric algorithm auto3dgm (Boyer et al., 2015a), a MATLAB application (available on GitHub). The algorithm is also available as an R-package, which can currently be downloaded at the following URL with documentation and tutorials (www.stat.duke.edu/~sayan/auto3dgm/index.shtml). Components of the method are detailed in Boyer et al. (2015a); here we present a brief explanation of the protocol. The analysis down-samples each surface to a uniform number of evenly spread landmarks—in this case 256 points, which it then uses to find pairwise alignments via the Closest Iterative Points algorithm (Besl & McKay, 1992). We reduce the risk of incorrect alignments by specifying eight initial alignments that represent all combinations of the first three principal axes of variation in the landmark points. 1,200 points were used to represent each bone’s surface. The initial set of pairwise distances between bones of the sample is used to define a minimum spanning tree linking all bones. Point correspondences are propagated through this network, allowing proper alignment of disparate shapes. This propagation process results in a final landmark dataset, and revised pairwise distance measures between all surfaces. We then used the Procrustes distance matrix in a Multidimensional Scaling Analysis with the MATLAB function ‘mds.m’ to condense the variation into two dimensions (the landmark output could also have been analyzed in morphologika2.5).

Phylogenetic analysis

We undertook multiple phylogenetic analyses to determine how the new character data from the Egerkingen tarsals influenced previous placements of Caenopithecus and other adapiforms. The phylogenetic analyses presented here build on a morphological character matrix that is based largely on the original work of Kay, Ross & Williams (1997), Ross, Williams & Kay (1998), Seiffert, Simons & Attia (2003) and Seiffert, Simons & Simons (2004), and which has been successively augmented by Seiffert et al. (2009), Seiffert et al. (2010) and Seiffert et al. (2005), Boyer, Seiffert & Simons (2010), Patel et al. (2012), Boyer & Seiffert (2013), Gladman et al. (2013), and, most recently, Boyer et al. (2015b). The matrix (Dataset S1) now includes 391 characters, and, with the addition of Mescalerolemur, a possible caenopithecine from the middle Eocene of Texas (Kirk & Williams, 2011), a total of 109 taxa. We undertook both parsimony and Bayesian analyses of this character matrix.

Two initial parsimony analyses were carried out using PAUP 4.10b10 (Swofford, 1998). For both, heuristic searches were run for 10,000 replicates with random addition sequence and the tree bisection and reconnection algorithm. For one of the two parsimony analyses, 256 characters whose states could be reasonably arranged into ordered (additive) morphoclines were treated as such. A subset of these ordered characters (209 total) had polymorphisms that were scored as intermediate states rather than scored using standard polymorphic scoring (i.e., (01)). These 209 characters were scaled so that transitions between “fixed” states were equal to a single step. In addition, we employed a molecular scaffold that constrained extant taxa to fit with the prevailing primate phylogeny based on molecular sequence data (specifically, the results of Springer et al. (2012)), and we constrained characters encoding premolar loss so that teeth that had previously been lost could not be regained. Another parsimony analysis was run with the molecular scaffold enforced, but with no assumptions about character ordering or premolar re-evolution—i.e., all characters were treated as unordered, with all transitions between states equal to a single step. Equally parsimonious trees recovered by these analyses are summarized here as strict consensus trees, and bootstrap support is provided, based on 1,000 pseudoreplicates (also calculated in PAUP).

Two Bayesian analyses were carried out using MrBayes 3.2.2 (Ronquist et al., 2012) and that program’s Mk model for morphological data. Both analyses were run on the CIPRES server (Miller, Pfeiffer & Schwartz, 2010). The same molecular scaffold as that used in the parsimony analyses was enforced in MrBayes using partial constraints. Analyses were run for 50 million generations, with four chains (three heated, one cold), sampling every 1,000 generations. Trees were summarized as a “halfcompat” consensus (50% majority-rule consensus) with a relative burn-in (25% of the samples). One analysis was run with all characters treated as unordered. While it would be ideal to run a Bayesian analysis of the matrix with all 256 characters treated as ordered as in the parsimony analysis described above, unfortunately MrBayes only allows multistate characters to be treated as ordered if they have six or fewer states, and 28 of the ordered characters in the parsimony analysis have >6 states. In order to run comparable parsimony and Bayesian analyses with all 256 characters treated as ordered, we removed the intermediate polymorphic states in the matrix and used standard polymorphic scoring. We consider this solution to be far from ideal, because it effectively renders those polymorphisms uninformative for phylogenetic reconstruction, but it is the only clear option that we could find for running comparable analyses while maintaining what we consider to be appropriate character state delimitations (the alternative being to collapse adjacent states into the same state, which would also lead to loss of information). In all of the Bayesian analyses coding was set to “variable” (lset coding = variable), which led to the exclusion of invariant or parsimony uninformative characters. The parsimony analysis of this modified matrix was run in the same way as the other parsimony analyses, as described above. We also run all of the same analyses, with all of the same assumptions, with the Egerkingen tarsals scored as belonging to Leptadapis priscus, the other large adapiform at Egerkingen fissure γ, in order to determine whether attribution to this taxon (rather than to Caenopithecus) has an impact on phylogenetic relationships among adapiforms.

Micro-CT scanning

The Egerkingen tarsals described here were micro-CT scanned at the American Museum of Natural History’s Microscopy and Imaging Facility, using a Phoenix brand v/tome/x s240 micro-CT scanner. High resolution scan and photographic imagery utilized here are available through MorphoSource.org. Scans of taxa used in comparative analyses and details on scanning facility, scanning resolution and energy settings are also largely available through MorphoSource.org. Additional details are available in appendix tables or supplementary information of Boyer et al. (2013) and Boyer & Seiffert (2013).

Results

Attribution

On the basis of astragalar trochlear width and calcaneo-cuboid joint surface size relative to M2 size, the possibility of the Egerkingen primate tarsals being attributable to either Anchomomys cf. pygmaeus or Necrolemur cf. zitteli (both of which have been recovered from Egerkingen γ) can be confidently excluded. Of the remaining possibilities (Caenopithecus lemuroides or the dentally smaller Leptadapis priscus) Stehlin (1916) lists 22 specimens in the Egerkingen γ collection of C. lemuroides (an additional six are known from the “Huppersand”), but only two specimens of L. priscus. Using a more conservative metric, the minimum number of C. lemuroides individuals represented in the Egerkingen γ collection is six, while the minimum number of L. priscus individuals represented in Egerkingen γ collection is two. On the basis of abundance, the most likely attribution is to C. lemuroides. This is particularly true when the number of isolated tarsals is taken in account—i.e., it is much more likely that four isolated tarsals (all likely attributable to a single species) would be derived from the species represented by 22 non-tarsal specimens (C. lemuroides), than to the species represented by only two non-tarsal specimens (L. priscus).

Regressions of M2 area against tarsal dimensions showed high coefficients of determination of 0.94 (trochlear width) and 0.93 (calcaneo-cuboid facet size) (Fig. 1). The M2 area of C. lemuroides is within the computed prediction intervals for M2 area generated by both tarsal elements, though it has a high positive residual indicating that, if the tarsals are attributable to C. lemuroides, that species would have relatively small tarsal facets compared to M2 area (Fig. 1). The M2 area of L. priscus also falls within this interval, but with a fairly small and slightly negative residual. On this basis, the tarsals could belong to either species and would apparently be more likely candidates for attribution to L. priscus.

Using a modified regression that excludes large adapiforms, we then compared typical residual values of these adapiforms to those of the focal fossils. We found that the residual values of the two candidate owners of the Egerkingen tarsals are both well outside of the 95% confidence limits on the means of the residual value distributions exhibited by other large adapiforms (Fig. 2). Computing the significance of the difference between the focal fossil residual values and the means for the known associations, we find that, for the astragalus, the probability of attribution to L. priscus is lower (t-test of null hypothesis that the adapiform mean is equal to that of L. priscus residual of −0.23; adapiform mean is 0.18, 95% C.I. is 0.05–0.30; t-value = 7.30; p (null correct) = 0.00003) than to C. lemuroides (residual of 0.51; t-value = −5.94; p (null correct) = 0.0001). For the calcaneus, the probability of attribution to C. lemuroides is slightly lower (t-test of null hypothesis that adapiform mean is equal to the C. lemuroides residual of 0.57; adapiform mean is 0.19, 95% C.I. is 0.04–0.33; t-value = −5.85; p (null correct) = 0.0004) than attribution to L. priscus (residual of −0.18; t-value 5.67; p (null correct) = 0.0005). However, the differences are minimal in both cases, and again we note that NMB Eh 719 and NMB En.270 have strange (but morphologically and metrically compatible) articular surfaces, arguing against attribution to two different species of different dental size. Nonetheless, these results may lead one to question whether the bones belong to two different taxa.

Figure 2 Residuals for the natural log of M2 area of adapiforms following recalculation of regression of the natural log of M2 area on tarsal facet area excluding those adapiform species.

Note that Caenopithecus lemuroides (“Cl”) and Leptadapis priscus (“Lp”) have predicted values for the natural log of lower second molar area that fall outside of the 95% confidence interval on the mean for both calcaneocuboid facet area (above) and trochlear width (below). Other abbreviations: “Ap,” Adapis parisiensis; “Al,” Afradapis longicristatus; “Ca,” Cantius abditus; “Cm,” Cantius mckennai; “Cr,” Cantius ralstoni; “Ct,” Cantius trigonodus; “Ek,” Europolemur klatti; “Nr,” Notharctus robustior; “Nt,” Notharctus tenebrosus; “Lm,” Leptadapis magnus; “Sg,” Smilodectes gracilis. See figure for regression equations.

The comparison of absolute values of z-score differences between calcaneal and astragalar regression residuals helps to address the last concern, though it cannot completely resolve it (Fig. 3). Computing this value for 58 taxa (i.e., all those species with data available for both bones in the original regression, including those of adapines), we find the average difference between z-score converted residuals is about 0.6 standard deviation units, with a standard deviation of 0.42 units, and a full range from 0.017–1.96. Whether both fossils are treated as C. lemuroides, or both are treated as L. priscus, the residual differences do not reject the hypothesis that these bones belong to a single taxon. It also leads to the expectation that if, in fact, one bone represents C. lemuroides and the other represents L. priscus, then these animals would have had very similarly-sized tarsals overall, despite smaller teeth in the latter.

Figure 3 Box plot of z-score-standardized differences in residuals.

Values plotted are the absolute difference between the z-score-standardized residual of m2 size to astragalus size and the residual of m2 size to calcaneus size of a given taxon. We noticed that for most taxa, the m2 residual generated by the calcaneus was proportional to the m2 residual generated by the astragalus. The comparative sample includes all specimens for which both astragali and calcanei could be compared with tooth size (both fossil and extant) for a total n = 57 species. As in other plots, the diamond enclosed cross uses the molar measurements of Caenopithecus, while the regular cross uses those of Leptadapis priscus. Because the fossils under scrutiny plot in the observed range, this test does not refute the hypothesis that both bones belong to a single species.

The last relevant observation emerging from these analyses is that the residual values of putative close relatives of Caenopithecus (Europolemur klatti (Thalmann, 1994) and Afradapis (Seiffert et al., 2009)) are not only positive, but are well above the general adapiform means (Fig. 2). The closest relatives of L. priscus in the sample are the adapines Adapis parisiensis and Leptadapis magnus. For the astragalus, both of those taxa still exhibit a positive residual in contrast to the L. priscus attribution, and A. parisiensis exhibits a value greater than the adapiform mean. For the calcaneus, Adapis also has a positive residual, though it is on the same side of the adapiform mean as the L. priscus residual. L. magnus has a slightly negative residual, putting it in closer proximity to the L. priscus residual. Higher than average positive residuals are therefore expected for caenopithecine tarsal-dental comparisons (meaning that caenopithecines are expected to have larger M2s relative to tarsal size than the average adapiform), while neutral to somewhat positive residuals (that are slightly below the average for adapiforms) are expected for adapine tarsal-dental comparisons. This makes sense given what is known about the folivorous caenopithecine Afradapis (Seiffert et al., 2009; Seiffert et al., 2010) because primate folivores are expected to have large postcanine teeth relative to body mass (e.g., Kay, 1975; Scott, 2011). Previous analyses of shearing quotients indicate that Caenopithecus was very likely folivorous (Kay et al., 2004), and we obtained the same result (Appendix S4, Fig. S1; see dental topographic variables for Caenopithecus in Table 3) using the dental topographic comparative framework employed by Seiffert et al. (2010) (though we note that Caenopithecus plots close to Prolemur simus, which was grouped with folivores in our analysis but is technically a bamboo specialist). As stated above, primate species with folivorous/fibrous diets are expected to have relatively large teeth, and thus large positive residuals of tooth size from tarsal size. Of the candidate species that the tarsals might belong to, C. lemuroides residuals meet this expectation, but L. priscus does not meet the expectation of neutral to slightly positive residuals based on the data available from other adapines.

Finally, on the basis of the overall morphological pattern, NMB En.270 is more similar to the astragalus of the caenopithecine Afradapis than it is to those of adapines. C. lemuroides has been placed as the sister taxon of Afradapis to the exclusion of all other living and extinct primates in the phylogenetic analysis of Seiffert et al. (2009) (and all later analyses that augmented that matrix), as well as in analysis of a larger character matrix that was independently constructed by Ni et al. (2013). More details on these similarities are presented below. The morphology of NMB En.270 is therefore certainly phylogenetically consistent with attribution to C. lemuroides, given what is known about its currently recognized sister taxon. L. priscus has been placed as the sister taxon to L. magnus and Adapis based on dental data (Boyer et al., 2015b), and the latter taxa also show a number of similar morphological features of the astragalus and calcaneus. If the Egerkingen tarsals belonged to L. priscus, we would expect the morphology to reflect an antecedent condition, or similar specializations, to those of Adapis or Leptadapis; instead the tarsal specimens exhibit some unusual specializations that are not expressed in these taxa.

In light of all the foregoing—i.e., on the combined basis of abundance, size (including the higher than average positive residuals for the Caenopithecus tarsal-dental comparisons, which fit expectations for a folivorous caenopithecine, but not an adapine), and morphology—we consider the most parsimonious attribution of the Egerkingen tarsals to be to C. lemuroides rather than L. priscus.

Body mass estimates

Using the prediction equations published by Yapuncich, Gladman & Boyer (2015) astragalar ectal facet area (AEFa) and calcaneal ectal facet area (CEFa) from NMB En.270 and NMB Eh 719, respectively, returned mean estimates of 1,663 g (AEFa; 95% PI = 659–4,196 g) and 2,217 g (CEFa; 95% PI = 1,098–4,476 g) using their “strepsirrhine” equation; 2,023 g (AEFa; 95% PI = 342–11,984 g) and 2,104 g (CEFa; 95% PI = 609–7,268 g) using their “lorisiform” equation; and 2,962 g (CEFa; 95% PI = 1,392–6,305 g) and 1,964 g (AEFa; 95% PI = 556–6,929 g) using their “lorisid” equation.

Description of the Caenopithecus tarsals

Astragalus (NMB En.270, Fig. 4). The astragalus is largely complete, but is abraded along the lateral trochlear rim, the most proximal and medial aspect of the body (the medial tubercle buttressing the groove for the flexor fibularis tendon), and probably (but less clearly) the most distal and lateral aspect of the ectal facet and fibular facets, where those two facets typically meet. The specimen is also missing most of the head and the navicular facet.

Figure 4 Stereopair images of NMB En.270, left astragalus from Egerkingen (probably fissure γ) attributed here to Caenopithecus lemuroides.

(A) proximal, (B) distal, (C) medial, (D) lateral, (E) dorsal, and (F) plantar views.

In medial view the astragalar body is dorsoplantarly tall, with a medial trochlear rim that has a small radius of curvature (i.e., it is tightly curved for its proximodistal length). The lateral tibial facet (trochlea) bears a shallow sulcus between the medial and lateral rims, which are of about the same height in proximal view. In dorsal view, the lateral tibial facet has a fairly straight medial border along the body (i.e., excluding the distal extension that is confluent with the medial tibial facet), but the lateral border is rounded and tapers strongly toward the proximal and medial aspect of the body; the lateral tibial facet is thus distinctly “v”-shaped in proximal view, and the proximal tapering of the facet allows for a capacious groove for the tendon of the flexor fibularis muscle, which is situated lateral to the lateral tibial facet, as in all known adapiforms and crown strepsirrhines (Beard et al., 1988). Unlike most other primates, this groove extends well onto the plantar aspect of the body and is buttressed laterally by a thick and laterally projecting flange, which also supports the most proximal aspect of the ectal facet, and presumably served as an attachment site for the posterior astragalo-fibular ligament. There is no hint of a posterior astragalar shelf or a superior astragalar foramen.

The fibular facet is proximodistally convex and quite large, covering most of the lateral aspect of the astragalar body in lateral view. The fibular facet slopes laterally from the lateral tibial facet at an angle of 112° (using the measurement protocol described by Boyer & Seiffert, 2013). The medial tibial facet is large, taking up more than half of the medial surface of the astragalar body and extending all the way to its plantar surface, as in many other “prosimian” primates (Boyer et al., 2015b). The facet continues distally, becoming dorsoplantarly shorter as it curves onto the medial surface of the astragalar neck. There is a dorsoplantarly tall but shallow bean-shaped fossa proximal to the medial tibial facet, presumably for attachment of the posterior tibioastragalar portion of the deltoid ligament. The articular area of the medial tibial facet measures 25.7 mm2. The medial tubercle buttressing the groove for the flexor fibularis tendon is abraded proximally, but the plantar border of the medial wall of the body clearly extends far plantarly as a protruding ridge, which terminates just distal to the most proximal extension of the sustentacular facet. This plantar projection forms the medial half of the deep trochlear groove for the flexor fibular tendon as it passes plantarly around the astragalus, and clearly contributes to the perception that the astragalar body is tall.

We estimate that the astragalar neck meets the body at approximately a 33° angle. Judging from the shape of the astragalar neck along its broken surface, the navicular facet likely would have been mediolaterally quite broad relative to its dorsoplantar height. A small part of the navicular facet is preserved on the lateral aspect of the neck, revealing that that part of the facet, at least, was clearly convex. On the plantar surface, the proximal part of the sustentacular facet is preserved, and is strikingly convex, with articular surface extending medially and laterally away from the plantar apex of the facet. There is no concave extension of the sustentacular facet along its proximal margin, as occurs in some primates. The facet’s proximolateral border is well-defined, but the proximomedial border is not, sloping gradually toward the sulcus that separates the facet from the plantar ridge on the body’s medial wall. An elongate tubercle is present on the dorsal surface of the neck, presumably for attachment of the astragalar-ectocuneiform ligament.

The ectal facet is roughly rectangular in plantar view, with a laterally projecting proximal part where the facet extends out onto the lateral tubercle for the flexor fibularis groove. The medial margin of the ectal facet is well-defined and projects plantarly, forming the lateral wall of the deep flexor fibularis groove. The area of the ectal facet is 20.6 mm2. See Table 2 for astragalar measurements taken on the specimen that follow the methods of Gebo et al. (2001).

Table 2 Measurements of the Egerkingen tarsals, following Gebo et al. (2001).

Measurement	Specimen	
	NMB En.268	NMB En.269	NMB Eh 719	
“Calcaneal length” (CalL)	22.25		20.74	
“Distal calcaneal length” (DistL)			7.76	
“Posterior calcaneal facet length” (PcfL, = ectal facet of this study)		7.54	6.87	
“Posterior calcaneal facet width” (PcfW)	2.61		3.77	
“Heel length” (HeelL)	8.98		6.24	
“Calcaneal width” (CalW)			10.11	
“Calcaneocuboid height” (CubHt)			4.17	
“Calcaneocuboid width” (CubW)			6.51	
CalW/CalL			0.49	
DistL/CalL			0.37	
PcfL/CalL			0.33	
HeelL/CalL	0.40		0.30	
PcfL/HeelL			1.10	
PcfW/PcfL			0.55	
CubW/CubHt			1.56	
	NMB En.270	
“Talar neck angle” (TNECKANGLE, in degrees, estimated)	33	
“Trochlear length” (TRL)	9.33	
“Midtrochlear width” (MTRW)	6.25	
“Talar width” (TW)	9.28	
MTRW/TRL	0.67	

Table 3 Relief index (RFI) and orientation patch count (OPC) values for P4 and M2 of Caenopithecus lemuroides.

Specimen	Species	Locus	2D area	3D area	RFI	OPC	
NMB no number	Caenopithecus lemuroides	p4	15.77	43.18	0.50	35.50	
NMB no number	Caenopithecus lemuroides	m2	25.96	67.60	0.48	68.00	
NMB Eh 396	Caenopithecus lemuroides	m2	25.89	67.24	0.48	54.13	
NMB Eh 735	Caenopithecus lemuroides	m2	26.05	71.03	0.50	62.63	
NMB Eh 600	Caenopithecus lemuroides	p4	16.10	38.86	0.44	28.63	
NMB Eh 597	Caenopithecus lemuroides	p4	14.59	41.40	0.52	36.88	
NMB Eh 597	Caenopithecus lemuroides	m2	25.24	68.45	0.50	56.25	

Calcanei (NMB Eh 719 (Fig. 5), NMB En.268 (Fig. 6), and NMB En.269 (Fig. 7)). Three primate calcanei have been identified in the Egerkingen collections. As already discussed, the three calcanei are of approximately the same size and, despite some differences, conform to a unique morphological pattern. Unless otherwise mentioned, the following description is based largely on NMB Eh 719, which is both the best-preserved calcaneus, and the only specimen that is known to be from fissure γ based on available records (Fig. 5). Of the other two calcanei, NMB En.268 (Fig. 6) is missing the medial part of the sustentaculum and has a large crack passing mediolaterally through the sustentaculum and ectal facet, leading to displacement of the two halves of the bone relative to each other; it is also missing most of the cortical bone on the lateral surface of the calcaneal tuber. The medial aspect of the sustentaculum of NMB En.269 (Fig. 7) is also missing, and the bone is badly abraded all along the medial and plantar surface, including the cuboid facet. In dorsal view, most of the lateral part of the ectal facet of NMB En.269 is missing, and the proximal aspect of the calcaneal tuber is badly damaged.

Figure 5 Stereopair images of NMB Eh 719, right calcaneus from Egerkingen fissure γ, attributed here to Caenopithecus lemuroides.

(A) dorsal, (B) distal, (C) lateral, (D) medial, and (E) plantar views.

Figure 6 Stereopair images of NMB En.268, right calcaneus from Egerkingen (probably fissure γ) attributed here to Caenopithecus lemuroides.

In (A) dorsal, (B) distal, (C) lateral, (D) medial, and (E) plantar views.

Figure 7 Stereopair images of NMB En.269, left calcaneus from Egerkingen (probably fissure γ) attributed here to Caenopithecus lemuroides.

In (A) dorsal, (B) distal, (C) lateral, (D) medial, and (E) plantar views.

The distal segment of NMB Eh 719 makes up approximately 37% of total calcaneal length and is not dorsally “flexed” relative to the proximal segment (see Gladman et al., 2013). The cuboid facet of NMB Eh 719 is damaged along its dorsomedial and plantar surface, but it is clear that the articular surface is “fan”-shaped, with a long axis that is oriented obliquely with respect to the dorso-plantar axis of the calcaneal body. There is a distinct concavity along the medial and plantar surface of the facet for articulation with a proximally projecting process of the cuboid. Medial and plantar to this articular pit is a proximodistally elongate distal calcaneal tubercle (best developed in NMB Eh 719 and NMB En.269). The cuboid facet and its margins are best preserved on NMB En.268, and this specimen confirms that the sustentacular facet does not extend to the distal end of the calcaneus, that there is no secondary sustentacular facet, and that there is no facet for the navicular distal to the sustentacular facet. At its broadest point (i.e., at the most medial projection of the sustentacular shelf), NMB Eh 719 is about 48% as wide as the calcaneus is long. The sustentaculum does not have a deep groove for the passage of the flexor fibularis tendon, but rather is quite flat—particularly that of NMB Eh 719, but somewhat less so in NMB En.268. This condition is surely correlated with the dorsolateral orientation of the entire sustentacular shelf, which is best appreciated in distal view (Figs. 5D, 6D and 7D). In dorsal view the sustentacular facet of NMB Eh 719 is proximodistally elongate, bean-shaped, and bears a gentle lateral concavity; it tapers distally and plantarly to meet the body of the calcaneus (the facet is, however, broken along its proximal and medial margin). The proximal margin of the sustentaculum does not bear a convex articular surface for a proximal extension of the astragalar sustentacular facet, as occurs in some other Paleogene primates. The ectal facet is about 55% as wide as it is long, and tapers proximally but remains quite broad distally. The facet is tightly curved in all specimens, though on NMB En.268 this condition is obscured somewhat by breakage and displacement of the distal part of the facet. The dorsal surface of the facet does not project out laterally above the lateral border of the calcaneus, as occurs in some primates that consequently bear a concave surface inferolateral to the ectal facet. The peroneal tubercle is placed at approximately the distal margin of the ectal facet, is longer (proximodistally) than it is high (dorsoplantarly), and does not project far laterally; it is not elongate and shelf-like as in some other Paleogene primates. The lateral wall of the distal segment in C. lemuroides appears to be medially oriented with respect to the proximal segment, but this might also be interpreted as a consequence of medial bowing of the calcaneal tuber with respect to the distal segment. The calcaneal tuber bears distinct rugosities not only along the dorsal surface but also along the medial margin, further contributing to its medially bowed appearance. See Table 2 for calcaneal measurements taken on the specimens that follow the methods of Gebo et al. (2001).

Comparisons with other strepsirrhine astragali

Given Caenopithecus’ well-supported phylogenetic placement among “adapiforms”—an assemblage of fossil primates that are basally diverging within the order and that may or may not be paraphyletic with respect to crown strepsirrhines (see phylogenetic results), we restrict our comparisons largely to these taxa, specifically adapines (Dagosto, 1983; Decker & Szalay, 1974), notharctines (Gebo, 1988; Gebo, Dagosto & Rose, 1991), asiadapines (Rose et al., 2009), Anchomomys (Moyà-Solà & Köhler, 1993; Moyà-Solà et al., 2011), Azibius (Marivaux et al., 2011), and Djebelemur (Marivaux et al., 2013). We also make comparisons based on figures of isolated specimens that have been attributed to Europolemur klatti (Thalmann, 1994) and figures and half-casts of specimens that have been attributed to Kyitchaungia takaii (Beard et al., 2007) from the middle Eocene of Germany and Burma, respectively. Beard et al. (2007) consider Kyitchaungia to be a sivaladapid; if correct, the specimens might be the only known tarsal elements from that clade (though we note that, on the basis of our comparisons of M2 size to tarsal dimensions, these specimens could also be attributable to the amphipithecid Myanmarpithecus yarshensis—a possibility that Beard et al. (2007) did not explicitly consider). Here we make comparisons with the astragalus (NMMP 59) and best-preserved calcaneus (NMMP 58) that Beard et al. (2007) attribute to Kyitchaungia. We also make comparisons with the distal calcaneus that is part of the problematic partial skeleton NMMP 20, from the late middle Eocene of Myanmar; this partial skeleton is either that of an amphipithecid, as originally suggested on the basis of the dental remains known from Sabapondaung kyitchaung locality (Ciochon et al., 2001), or is a sivaladapid (dental remains of which have not been found at the locality; Beard et al., 2007). In addition, we discuss similarities that the Caenopithecus tarsals share with those of lorisids, the subfossil indrioid Babakotia, and some other extant strepsirrhines, largely because of their importance for functional interpretation. Note that the same comparisons would also be appropriate if the Egerkingen tarsals actually belong to L. priscus.

Caenopithecus shows a unique mix of astragalar features, some of which are seen in Adapis and Afradapis, and others of which are more similar to conditions seen in Leptadapis and notharctines. The astragalar body is relatively tall (dorsoplantarly) when compared with those of Adapis (Fig. 8E), Afradapis (Fig. 8G), lorises (Figs. 9E and 9F), and Babakotia (Fig. 9G). The plantar aspect of the proximal portion of the astragalar body is not preserved in Adapoides (Fig. 8H), but the remaining morphology suggests that the astragalar body was probably fairly low, perhaps as in Afradapis. Among other stem strepsirrhines, relatively tall astragalar bodies are also seen in Leptadapis (Fig. 8D), notharctines such as Cantius (Fig. 8B), asiadapines (Fig. 8C), Anchomomys, Azibius, Djebelemur (Fig. 8A), and NMMP 59. The height of Babakotia’s astragalar body in medial view is exaggerated by its tall plantarly-projecting medial process buttressing the groove for the tendon of flexor fibularis (Fig. 9G).

Figure 8 Astragali of other fossil strepsirrhines compared to NMB En.270, attributed here to Caenopithecus lemuroides.

(A) Djebelemur martinezi (CBI-1-545), from the early or middle Eocene of Tunisia; (B) Cantius trigonodus (USGS 21832), from the early Eocene of the USA; (C) Asiadapis cambayensis (GU 747), from the early Eocene of India; (D) Leptadapis magnus (MNHN QU 11001), from the late Eocene of France; (E) Adapis parisiensis (ECA 1379), from the late Eocene of France; (F) Caenopithecus lemuroides (NMB En.270, reversed); (G) Afradapis longicristatus (DPC 21445C), from the late Eocene of Egypt; (H) Adapoides troglodytes (IVPP V13016, reversed), from the middle Eocene of China. Views in, from left to right, dorsal, plantar, proximal, lateral, distal, and medial views. Scale bars are equal to 1 mm.

Figure 9 Astragali of extant and subfossil strepsirrhines compared to NMB En.270, attributed here to Caenopithecus lemuroides.

(A) Microcebus murinus (AMNH 174430); (B) Hapalemur griseus (AMNH 170680); (C) Daubentonia madagascariensis (AMNH 119694); (D) Varecia variegata (AMNH 201384); (E) Perodicticus potto (AMNH 184579); (F) Nycticebus coucang (AMNH 212953); (G) Babakotia radofilai (DPC 11000); (H) Caenopithecus lemuroides (NMB En.270, reversed). Views in, from left to right, dorsal, plantar, proximal, lateral, distal, and medial views. Scale bars are equal to 1 mm.

The very well-defined triangular proximal extension of the lateral tibial facet is most like those of Afradapis and Babakotia (Figs. 8G and 9H). Other living and extinct strepsirrhines have facets that taper posteriorly, but they are not so distinctly set off from the flexor fibularis groove. In Caenopithecus that groove is deeply excavated and extends onto the plantar aspect of the body, as in adapines (Fugs. 8D and 8E), Afradapis (Fig. 8G), Babakotia (Fig. 9G), lorises (Figs. 9E and 9F), and, to a lesser extent, Varecia (Fig. 9D). In Afradapis, Caenopithecus, Babakotia, and some lorises, the proximomedial margin of the ectal facet projects distinctly plantar to the groove, forming its lateral wall. The plantar ridge forming the medial wall of this groove in Caenopithecus is also well-developed in Leptadapis (see MNHN QU 11001, Fig. 8D) and Babakotia, but is not as distinct in Adapis and Afradapis. This plantar projection clearly contributes to the perception that Caenopithecus and Leptadapis have tall astragalar bodies, but in medial view the neck and medial tibial facet of Caenopithecus are actually more similar to those of Afradapis, aside from the strongly projecting plantar ridge.

The medial tibial facet of Caenopithecus is dorsoplantarly deep as in all other adapiforms, but specifically resembles that of Afradapis in becoming dorsoplantarly shorter as it curves onto the medial surface of the astragalar neck; this similarity is surely also due to the relatively long astragalar necks of Afradapis and Caenopithecus when compared to those of adapines. The ln of the square root of medial tibial facet area relative to ectal facet area is 0.11, which is higher than that of any Paleogene primate. The only extant and subfossil primates that equal or exceed this value (i.e., that have an equal, or higher, ratio of medial tibial facet area relative to ectal facet area) are indriids, some lemurids, galagids, some lorisids (Arctocebus and Nycticebus), and Palaeopropithecus.

The complete absence of a posterior trochlear shelf also characterizes Adapis, Adapoides, Afradapis, Babakotia, and lorises; such “shelves” are present, to varying degrees, in notharctines, Anchomomys, Djebelemur, and some extant strepsirrhines, and Beard et al. (2007) suggest that one was probably present on NMMP 59. In medial view Leptadapis appears to bear a posterior trochlear shelf, but this posterior bulge is composed solely of the plantarly projecting ridge buttressing the flexor fibular groove, and there is no shelf extending across the proximal and plantar aspect of the body as in some other adapiforms and crown strepsirrhines. The absence of a superior astragalar foramen differs from the condition in Afradapis, Leptadapis, and some specimens of Adapis; this foramen also occurs variably in lorisids, but not in any other extant primates of which we are aware.

The fibular facet angle of 112° is slightly higher than those that have been calculated for Leptadapis (104–109°, n = 3), but is within the range of Adapis (106–114°, n = 8) and Babakotia (98–125°, n = 3), close to that of Djebelemur (113°), and lower than that of the single Afradapis specimen that is currently known (116°) (Boyer & Seiffert, 2013). Other adapiforms have lower values, for instance basal Cantius (94°−103°), Pelycodus (96°−108°), Asiadapis (100°), Marcgodinotius (106°−110°), and Anchomomys (105–111°).

The astragalar neck meets the body at approximately a 33° angle, similar to that of Afradapis and the values reported by Gebo et al. (2001) for Adapis and Leptadapis, but higher than the values that they reported for notharctines aside from Notharctus tenebrosus (35°). Among smaller stem strepsirrhines, astragalar neck angle is also relatively low—29° in Asiadapis, 20–32° in Marcgodinotius, 19° in Anchomomys, 17–18° in Azibius, 20° in Djebelemur, and 26° in NMMP 59 . The possible Europolemur klatti specimen (CeIV-2852) also appears to have a low neck angle, certainly lower than those of Afradapis or Caenopithecus. Lorises and Babakotia have particularly high neck angles.

The elongate tubercle on the dorsal surface of the neck for the astragalar-ectocuneiform ligament is also well-developed in Adapoides, Afradapis, Babakotia, lorises, many notharctines, and apparently on Europolemur, but is not clearly expressed to the same degree in adapines.

In many omomyiforms, asiadapines, adapines, and Adapoides there is a concave proximal extension of the sustentacular facet that would form a sort of locking mechanism with a posteriorly convex proximal extension of the sustentacular facet on the calcaneus (Boyer, Seiffert & Simons, 2010); in contrast, there is no distinct articular surface for such a proximal extension in either Caenopithecus or Afradapis. Caenopithecus is fairly unique among adapiforms in having a strongly mediolaterally convex proximal portion of the sustentacular facet.

Multivariate morphometric analysis of primate astragali

Along principal component one (which explains 39.4% of the variance) Caenopithecus falls close to Afradapis, and, among extant primates sampled, overlaps solely with the morphospace occupied by lorisids (Fig. 10). Among non-primates, Caenopithecus’ PC1 score is close to that of some Cynocephalus individuals, as well as the extinct carpolestid Carpolestes and paromomyid Ignacius. The strongest loadings along this axis are for variables 12 (flexor fibularis groove width), 20 (angle between fibular facet and medial tibial facet), and 21 (angle between medial and lateral tibial facets), all of which are positive. The positions of Caenopithecus, Afradapis, adapines, Babakotia, and lorisids along PC1 reflect the fact that, relative to other crown primates, they have broad flexor fibularis grooves, high (obtuse) angles between the fibular facet and the medial tibial facet, and low (acute) angles between the medial and lateral tibial facets. Principal component 2 explains 17.7% of the variance and does not clearly separate extant primates along functional lines; this component is dominated by a strong positive loading for variable 13 (flexor fibularis groove proximodistal length). Again, Caenopithecus overlaps with the adapine-caenopithecine and lorisid morphospaces along this axis (but also that of many other primates and non-primates). Overall, among extant primates that might guide functional interpretation, the clearest phenetic affinities along the principal two axes are with the cautious and slow-climbing lorisids. Importantly, though, on the far negative range of the lorisid morphospace there is near-overlap with the extant large-bodied acrobatic leaper Propithecus, presumably reflecting the fact that the reduced sample of variables that can be measured on the Caenopithecus astragalus does not adequately capture all of the functionally-informative morphology provided by this element (though we note that Propithecus is also capable of hind limb suspension in addition to acrobatic leaping).

Figure 10 Principal components analysis of astragalar shape variables and angles.

First two principal component axes, accounting for 57.1% of the overall variance, based on the reduced dataset of 15 astragalar measurements that could be taken on NMB En.270 (loadings for each variable on PC1 and PC2 are provided in the lower right hand corner). Note that Caenopithecus falls close to Afradapis, and to adapines and lorisids. The suspensory subfossil palaeopropithecid “sloth lemur” Babakotia expands the crown primate morphospace considerably, driven largely by extreme expression of features that also influence Caenopithecus’s positive score on PC1—i.e., a particularly long and wide flexor fibularis groove (variables 12 and 13) and a high angle between the medial tibial facet and fibular facet (variable 20).

Flexor fibularis groove depth

Quantification of flexor fibularis groove depth among living and extinct strepsirrhines revealed that most extant species have ratios of flexor fibularis groove contour length to flexor fibularis groove linear width of 1 or only slightly higher, indicating that there is very little concavity of the groove along the plantar surface of the astragalus (Fig. 11 and Table 4). This is universally characteristic of the particularly acrobatic grasp-leaping strepsirrhines, such as the galagids Galagoides and Otolemur, the lemurid Hapalemur, the lepilemurid Lepilemur, and the indriids Indri and Propithecus, but is also seen in extant lemurs with more generalized locomotor behavior. The major departures from ratios of 1 are seen in Cheirogaleus, Varecia, and particularly lorises, which have some of the highest values among extant taxa (of which Nycticebus and Perodicticus have the highest). The highest values among strepsirrhines, however, were found among the subfossil forms Babakotia (ratio of 1.73) and Megaladapis (mean ratio of 2.05). Archaeolemur also had a relatively high value when compared with those of most extant lemurs (mean of 1.1, close to that of Varecia). Among Eocene adapiforms, the phylogenetically basal taxa Asiadapis, Marcgodinotius, and Notharctus had values close to 1, while Afradapis and Caenopithecus both had values of 1.22, which is higher than the values of all extant strepsirrhines aside from those of some lorises (and note that the value for Caenopithecus is a minimum estimate, due to damage to the medial plantar ridge buttressing the groove). The adapines Adapis and Leptadapis had intermediate values, with means of 1.05 and 1.10, respectively.

Figure 11 Quantification of flexor fibularis groove depth on the plantar surface of the astragalus.

Inset image shows the orientation of the astragalus for simultaneous measurement of linear width of the flexor fibularis groove, and the contour measure of the groove (taken in Geomagic). As noted in the main text, astragali were oriented with the plantar surface facing upward and in posterior view, such that the point marking the base of the trough of the flexor fibularis groove was aligned with the point marking the plantar apex of the navicular facet or sustentacular facet (whichever was visible in that view). Boxplots show variation within species in the ratio of the contour measurement to the linear measurement; higher numbers are found in taxa with deeper flexor fibularis grooves. Note that the values for Babakotia and Megaladapis are so extreme that they fall far outside of the figured range.

Table 4 Ratios of flexor fibularis groove contour measures versus flexor fibularis groove linear width in living and extinct strepsirrhines.

Taxon	Specimen	Flexor fibularis groove contour/flexor fibularis groove linear	
Adapis parisiensis	MaPhQ 1390	1.03	
Adapis parisiensis	ROS 106	1.05	
Adapis parisiensis	ROS 2708	1.09	
Afradapis longicristatus	DPC 21445C	1.22	
Archaeolemur sp.	DPC 7849	1.08	
Archaeolemur sp.	DPC 7900	1.12	
Arctocebus calabarensis	AMNH 207949	1.11	
Arctocebus calabarensis	AMNH 212576	1.13	
Asiadapis cambayensis	GU 747	1.01	
Babakotia radofilai	DPC 11000	1.73	
Caenopithecus lemuroides	NMB En.270	1.22	
Cheirogaleus medius	DPC 0142	1.10	
Cheirogaleus medius	DPC 031	1.10	
Cheirogaleus medius	DPC 1023	1.10	
Daubentonia madagascariensis	AMNH 119694	1.03	
Eulemur fulvus albifrons	AMNH 170708	1.03	
Eulemur fulvus albifrons	AMNH 170728	1.04	
Eulemur fulvus fulvus	AMNH 31254	1.02	
Galagoides demidoff	AMNH 212956	1.01	
Galagoides demidoff	AMNH 241121	1.01	
Galagoides demidoff	AMNH 215180	1.01	
Hapalemur griseus	AMNH 170680	1.02	
Hapalemur griseus	AMNH 170689	1.04	
Hapalemur griseus	AMNH 61589	1.05	
Indri indri	AMNH 208992	1.00	
Indri indri	AMNH 100504	1.00	
Lemur catta	AMNH 170739	1.02	
Lemur catta	AMNH 170740	1.04	
Lemur catta	AMNH 170765	1.01	
Lepilemur mustelinus	AMNH 170556	1.01	
Lepilemur mustelinus	AMNH 170560	1.02	
Lepilemur mustelinus	AMNH 170565	1.00	
Leptadapis magnus	NMB QE 261	1.11	
Leptadapis magnus	NMB QE 496	1.09	
Loris tardigradus	AMNH 150038	1.24	
Loris tardigradus	AMNH 165931	1.10	
Loris tardigradus	AMNH 34257	1.11	
Marcgodinotius indicus	GU 748	1.02	
Marcgodinotius indicus	GU 749	1.01	
Megaladapis madagascariensis	DPC 18936	1.92	
Megaladapis madagascariensis	DPC 17176	1.81	
Megaladapis madagascariensis	DPC 7821	2.43	
Microcebus murinus	AMNH 174428	1.03	
Microcebus murinus	AMNH 174430	1.04	
Microcebus murinus	AMNH 174431	1.03	
Mirza coquereli	DPC 0137	1.00	
Mirza coquereli	DPC 1139	1.00	
Notharctus sp.	AMNH 12000	1.01	
Notharctus sp.	AMNH 11474	1.00	
Notharctus sp.	AMNH 129382	1.01	
Nycticebus coucang	AMNH 90381	1.29	
Nycticebus coucang	AMNH 102027	1.25	
Nycticebus coucang	AMNH 212953	1.15	
Otolemur crassicaudatus	AMNH 187364	1.00	
Otolemur crassicaudatus	AMNH 150413	1.01	
Otolemur crassicaudatus	AMNH 216240	1.00	
Perodicticus potto	AMNH 184579	1.23	
Perodicticus potto	AMNH 269851	1.11	
Perodicticus potto	AMNH 86898	1.30	
Propithecus verreauxi	AMNH 170474	1.00	
Propithecus verreauxi	AMNH 170463	1.01	
Propithecus verreauxi	AMNH 208991	1.00	
Varecia variegata	AMNH 201384	1.11	
Varecia variegata	DPC 049	1.12	
Varecia variegata	AMNH 100512	1.09	

Comparisons with other strepsirrhine calcanei

In terms of overall morphology and proportions, the calcanei of C. lemuroides are similar to those of Adapis and Leptadapis in having mediolaterally broad ectal facets (relative to proximodistal length), well-developed distal calcaneal tubercles, and “fan”-shaped (rather than strictly ovoid) facets for the articulating cuboid. This gestalt similarity to adapines is further supported by automated geometric analysis of calcanei from multiple living and extinct primates (see below). The C. lemuroides calcanei also show some striking specializations that, as a complex, clearly set them apart not only from adapines but all other adapiforms—including features such as tightly curved ectal facets that project dorsal to the calcaneal tubers, a convex lateral border of the calcaneus, and sustentacular shelves that are dorsolaterally inclined relative to the mediolateral plane of the ectal facets’ dorsal apices. Outside of adapiforms, some of these features can be found among lorises and Babakotia, and these probable convergences help to guide our functional inferences.

Relative elongation of the distal calcaneal segment in Caenopithecus (37% of total calcaneal length in NMB Eh 719) is intermediate between the very foreshortened condition exhibited by adapines (27–34%) and the values seen in the relatively elongate notharctines (38–45%), asiadapines (39–44%), and NMMP 58 (48%, Beard et al. (2007)); Anchomomys is unique among stem strepsirrhines in having even longer distal segments (51–53%). The value for NMB Eh 719 matches that which was reported for Europolemur by Gebo et al. (2001). The variable “ResB” of Boyer et al. (2013), which is the residual from a regression of absolute distal calcaneal segment length on estimated body mass across primates, is −0.16. Among Paleogene primates, this residual is higher than those exhibited by asiadapines and adapines, but lower than those of notharctines, Anchomomys, and all omomyiforms. Caenopithecus’ value is lower than those of all extant lemuriforms and galagids, but higher than those of lorisids (see Table 1 of Boyer et al., 2013). Among subfossil lemurs, Mesopropithecus has a much higher “Res B” residual, Megaladapis and Pachylemur have slightly higher residuals, while Archaeolemur, Babakotia, and Palaeopropithecus have much lower residuals.

The position of the peroneal tubercle in Caenopithecus is also intermediate between the relative placements in adapines and other adapiforms, although Asiadapis shows a similar pattern (Fig. 12A and Table 5). Europolemur klatti (GMH XXXII-196) has a somewhat proximally-placed peroneal tubercle—clearly proximal to the distal terminus of the ectal facet, and therefore more adapine-like than Caenopithecus. NMMP 20 and NMMP 58 have also been interpreted as having adapine-like positions of the peroneal tubercle (Ciochon et al., 2001; Beard et al., 2007). The peroneal tubercles of adapines are particularly well-developed and protrude markedly directly plantar to the apex of the ectal facet; they are so different from those of other adapiforms in their robust construction that Decker & Szalay (1974) even questioned their homology with those of their relatives. In strong contrast, the peroneal tubercles of Caenopithecus have proximal and distal borders that grade gradually into the body of the calcaneus.

Figure 12 Peroneal tubercle position in living and extinct primates, and comparisons of the Caenopithecus calcaneus NMB Eh 719 with those of other adapiforms.

(A) Box and whisker plots of ((natural log of proximal segment length)—(natural log of the position of peroneal tubercle midpoint)) measured on the calcanei of 50 Eocene adapiforms, 21 Eocene omomyiforms, four Eocene stem anthropoids, five Oligocene parapithecids, five Paleocene plesiadapiforms, and 15 extant non-primate euarchontans. Note that the range of Caenopithecus is intermediate between those of adapines and other adapiforms, but the broad range is largely driven by a single specimen. (B) Calcanei of various adapiforms scaled to the same approximate proximal segment length, illustrating differences in peroneal tubercle position (peroneal tubercles are delimited by opaque overlays). Also note the very short distal calcaneal segments of Adapis and Leptadapis relative to those of Caenopithecus and other adapiforms, and the different shapes and orientations of the long axes of the cuboid facets in distal view (margins of the cuboid facets are also delimited by opaque overlays). Scale bar = 10 mm.

Table 5 Measurements of peroneal tubercle position and size taken on calcanei of living and extinct euarchontans.

Higher taxon	Genus/species	Specimen	(1) ProxL	(2) DistPT	(3) MidPT	(4) (3–1)	(5) (2–1)	(6) [(2–3)–1]	
Adapinae	Adapis parisiensis	NMB QE 530	14.74	11.82	9.95	−0.393	−0.221	−2.065	
Adapinae	Adapis parisiensis	NMB QE 644	12.33	9.62	8.22	−0.405	−0.248	−2.176	
Adapinae	Adapis parisiensis	NMB QE 741	11.95	10.25	7.87	−0.418	−0.153	−1.614	
Adapinae	Adapis parisiensis	NMB QE 779	13.3	12.05	9.72	−0.314	−0.099	−1.742	
Adapinae	Adapis parisiensis	NMB QF 558	10.88	8.02	7.39	−0.387	−0.305	−2.849	
Adapinae	Adapis parisiensis	NMB QH 640	12.81	10.55	9.57	−0.292	−0.194	−2.570	
Adapinae	Leptadapis magnus	NMB QW 1676	22.56	19.7	16.55	−0.310	−0.136	−1.969	
Adapinae	Leptadapis magnus	PQ 1746	23.49	19.58	15.8	−0.397	−0.182	−1.827	
Adapinae	Leptadapis magnus	ACQ 266	24.74	22.05	16.95	−0.378	−0.115	−1.579	
Adapinae	Leptadapis magnus	ACQ 267	24.63	22.09	17.97	−0.315	−0.109	−1.788	
Adapinae	Leptadapis magnus	NMB QE 920	24.56	19.94	17.4	−0.345	−0.208	−2.269	
Adapinae	Leptadapis magnus	NMB QE 604	21.13	17.9	15.61	−0.303	−0.166	−2.222	
Adapinae	Leptadapis magnus	NMB QF 421	23.15	19.02	16.73	−0.325	−0.197	−2.313	
Adapinae	Leptadapis magnus	NMB QE 830	21.61	18.6	16.15	−0.291	−0.150	−2.177	
Asiadapinae	Asiadapis cambayensis	GU 716	5.76	5.89	na	na	0.022	na	
Asiadapinae	Asiadapis cambayensis	GU 760	6.64	6.71	6.01	−0.100	0.010	−2.250	
Asiadapinae	Marcgodinotius indicus	GU 1644	4.19	4.98	4.13	−0.014	0.173	−1.595	
Asiadapinae	Marcgodinotius indicus	GU 709	4.55	5.33	4.52	−0.007	0.158	−1.726	
Asiadapinae	Marcgodinotius indicus	GU 751	4.82	5.71	5.06	0.049	0.169	−2.004	
Asiadapinae	Marcgodinotius indicus	GU 1642	4.59	4.9	4.42	−0.038	0.065	−2.258	
Asiadapinae	Marcgodinotius indicus	GU 1643	4.33	5.21	4.39	0.014	0.185	−1.664	
Caenopithecinae	Caenopithecus lemuroides	NMB Eh 719	13.12	14.39	12.58	−0.042	0.092	−1.981	
Caenopithecinae	Caenopithecus lemuroides	NMB En.268	14.12	14.46	11.61	−0.196	0.024	−1.600	
Caenopithecinae	Caenopithecus lemuroides	NMB En.269	13.76	15.31	13.27	−0.036	0.107	−1.909	
Carpolestidae	Carpolestes simpsoni	UM 101963	4.12	6.85	5.89	0.357	0.508	−1.457	
Cynocephalidae	Cynocephalus volans	UNSM 11501	7.92	12.37	10.25	0.258	0.446	−1.318	
Cynocephalidae	Cynocephalus volans	AMNH 207001	9.08	14.37	12.3	0.304	0.459	−1.479	
Cynocephalidae	Galeopterus variegatus	USNM 317118	7.5	13.25	11.52	0.429	0.569	−1.467	
Eosimiidae	Eosimias sp.	IVPP 11851	3.59	4.51	3.84	0.067	0.228	−1.679	
Eosimiidae	Eosimias sp.	IVPP 12313	4.33	5.98	4.62	0.065	0.323	−1.158	
Eosimiidae	Eosimias sp.	IVPP 12281	4.08	4.73	3.67	−0.106	0.148	−1.348	
Notharctinae	Cantius mckennai	USGS 5897	10.45	12.72	11.4	0.087	0.197	−2.069	
Notharctinae	Cantius sp.	USGS 6791	11.76	12.89	11.71	−0.004	0.092	−2.299	
Notharctinae	Cantius sp.	USGS 21768	13.96	16.13	14.2	0.017	0.144	−1.979	
Notharctinae	Cantius abditus	USGS 21771	14.04	16.91	15.23	0.081	0.186	−2.123	
Notharctinae	Cantius abditus	USGS 21774	13.52	15.12	13.41	−0.008	0.112	−2.068	
Notharctinae	Cantius sp.	USGS 21778	12.24	13.98	12.71	0.038	0.133	−2.266	
Notharctinae	Cantius abditus	USGS 21825	13.53	15.06	13.66	0.010	0.107	−2.268	
Notharctinae	Cantius abditus	USGS 21827	14.25	16.34	13.99	−0.018	0.137	−1.802	
Notharctinae	Cantius frugivorus	USGS 21828	11.28	12.69	11.11	−0.015	0.118	−1.966	
Notharctinae	Cantius mckennai	USGS 25029a	11.28	13.25	12.04	0.065	0.161	−2.232	
Notharctinae	Cantius mckennai	USGS 25029b	11.28	13.1	11.73	0.039	0.150	−2.108	
Notharctinae	Cantius abditus	AMNH 16852	13.05	15.09	13.75	0.052	0.145	−2.276	
Notharctinae	Cantius abditus	USGS 6783	13.67	15.95	14.06	0.028	0.154	−1.979	
Notharctinae	Cantius ralstoni	UF 252980	8.63	11.12	10.43	0.189	0.254	−2.526	
Notharctinae	Cantius sp.	USGS 21829	12.41	14.61	13.41	0.077	0.163	−2.336	
Notharctinae	Cantius trigonodus	USGS 6774	14.98	15.21	14.12	−0.059	0.015	−2.621	
Notharctinae	Notharctus tenebrosus	AMNH 13766	15.46	18.42	16.35	0.056	0.175	−2.011	
Notharctinae	Notharctus tenebrosus	AMNH 55061	13.27	16.17	13.65	0.028	0.198	−1.661	
Notharctinae	Notharctus tenebrosus	AMNH 129382	13.56	16.07	14.28	0.052	0.170	−2.025	
Notharctinae	Notharctus tenebrosus	AMNH 11474	13.25	15.01	13.46	0.016	0.125	−2.146	
Notharctinae	Notharctus tenebrosus	AMNH 131945	14.26	17.66	15.05	0.054	0.214	−1.698	
Notharctinae	Notharctus tenebrosus	AMNH 131955	14.79	18.07	16.53	0.111	0.200	−2.262	
Notharctinae	Notharctus tenebrosus	AMNH 11478	15.57	18.57	16.73	0.072	0.176	−2.136	
Notharctinae	Smilodectes gracilis	USNM 21815	14.5	16.85	14.95	0.031	0.150	−2.032	
Notharctinae	Smilodectes gracilis	USNM 25686	14.76	16.38	14.82	0.004	0.104	−2.247	
Notharctinae	Smilodectes gracilis	AMNH 131763	15.02	17.46	15.45	0.028	0.151	−2.011	
Notharctinae	Smilodectes gracilis	AMNH 131774	14.85	16.96	14.93	0.005	0.133	−1.990	
Omomyiformes	Arapahovius gazini	UCMP 118498	3.59	4.33	3.9	0.083	0.187	−2.122	
Omomyiformes	Arapahovius gazini	UCMP 118499	3.39	4.43	3.83	0.122	0.268	−1.732	
Omomyiformes	Arapahovius gazini	UCMP 173038	3.79	4.76	4.21	0.105	0.228	−1.930	
Omomyiformes	Hemiacodon gracilis	AMNH 12613	7.61	8.01	6.34	−0.183	0.051	−1.517	
Omomyiformes	Omomys carteri	UCM 67678	6.93	7.79	6.52	−0.061	0.117	−1.697	
Omomyiformes	Omomys carteri	UCM 68745	7.23	8.61	6.97	−0.037	0.175	−1.484	
Omomyiformes	Omomys carteri	UCM 69065	7.52	9.69	7.13	−0.053	0.254	−1.078	
Omomyiformes	Omomys carteri	UCM 67679	7.04	8.82	6.53	−0.075	0.225	−1.123	
Omomyiformes	Omomys carteri	UCM 69303	7.88	9.08	8.14	0.032	0.142	−2.126	
Omomyiformes	Omomys carteri	UM 98604	6.45	7.01	6.4	−0.008	0.083	−2.358	
Omomyiformes	Ourayia uintensis	SDNM 4020-60933	10.96	12.58	8.77	−0.223	0.138	−1.057	
Omomyiformes	Shoshonius cooperi	CM 69765	4.43	5.3	4.49	0.013	0.179	−1.699	
Omomyiformes	Teilhardina belgica	IRSNB M1236	3.49	4.99	4.03	0.144	0.358	−1.291	
Omomyiformes	Teilhardina belgica	IRSNB M1237	3.09	4.2	3.63	0.161	0.307	−1.690	
Omomyiformes	Teilhardina belgica	IRSNB M1247	3.2	4.47	3.9	0.198	0.334	−1.725	
Omomyiformes	Teilhardina belgica	IRSNB 16786-06	3.27	4.28	3.67	0.115	0.269	−1.679	
Omomyiformes	Teilhardina belgica	IRSNB 26857-05	3.19	4.12	3.62	0.126	0.256	−1.853	
Omomyiformes	Teilhardina belgica	IRSNB 26857-04	3.48	4.77	4.15	0.176	0.315	−1.725	
Omomyiformes	Teilhardina belgica	IRSNB M0061	3.13	4.35	3.62	0.145	0.329	−1.456	
Omomyiformes	Teilhardina belgica	IRSNB 16786-03	3.31	4.63	4.01	0.192	0.336	−1.675	
Omomyiformes	Tetonius sp.	AMNH 88821	4.49	6.7	4.49	0.000	0.400	−0.709	
Parapithecidae	Apidium phiomense	DPC 1003B	14.12	17.39	15.22	0.075	0.208	−1.873	
Parapithecidae	Apidium phiomense?	DPC 8810	13.45	17.52	14.81	0.096	0.264	−1.602	
Parapithecidae	Apidium phiomense	DPC 2381	13.79	17.41	15	0.084	0.233	−1.744	
Parapithecidae	Apidium phiomense	DPC 15679	14.06	17.4	15.05	0.068	0.213	−1.789	
Parapithecidae	Apidium phiomense	DPC 20576	9.72	12.4	10.45	0.072	0.244	−1.606	
Paromomyidae	Ignacius graybullianus	USNM 442240	4.82	7.04	6.02	0.222	0.379	−1.553	
Plesiadapidae	Nannodectes gidleyi	AMNH 17379	6.49	8.95	7.48	0.142	0.321	−1.485	
Plesiadapidae	Plesiadapis churchilli	UM no number	7.71	10.53	9.01	0.156	0.312	−1.624	
Plesiadapidae	Plesiadapis cookei	UM 87990	11.13	16.05	12.91	0.148	0.366	−1.265	
Plesiadapidae	Plesiadapis tricuspidens	MNHN R 414	12.25	17.23	14.65	0.179	0.341	−1.558	
Proteopithecidae	Proteopithecus sylviae	DPC 24776	7.19	8.45	7.74	0.074	0.161	−2.315	
Ptilocercidae	Ptilocercus lowii	USNM 488067	3.13	3.74	3.05	−0.026	0.178	−1.512	
Ptilocercidae	Ptilocercus lowii	USNM 488069	3.13	3.73	3.03	−0.032	0.175	−1.498	
Ptilocercidae	Ptilocercus lowii	USNM 488072	3.13	3.66	3.05	−0.026	0.156	−1.635	
Tupaiidae	Tupaia belangeri	AMNH 113135	5.3	7.16	5.51	0.039	0.301	−1.167	
Tupaiidae	Tupaia glis	SBU Tg01	5.09	7.72	5.26	0.033	0.417	−0.727	
Tupaiidae	Tupaia glis	AMNH 215175	4.9	7.58	5.63	0.139	0.436	−0.921	
Tupaiidae	Tupaia glis	AMNH 215176	4.62	6.96	4.91	0.061	0.410	−0.813	
Tupaiidae	Tupaia glis	AMNH 215177	5.23	7.9	5.92	0.124	0.412	−0.971	
Tupaiidae	Tupaia glis	AMNH 215178	5.18	7.63	5.91	0.132	0.387	−1.102	
Tupaiidae	Tupaia glis	AMNH 215179	4.99	6.93	5.41	0.081	0.328	−1.189	
Notes.

ProxL length of the proximal segment

DistPT distance from the proximal-most aspect of the calcaneal tuber to the distal-most projection of the peroneal tubercle

MidPT distance from the proximal-most aspect of the calcaneal tuber to the midpoint of the peroneal tubercle

(3-1) MidPT—ProxL

(2-1) DistPT—ProxL

(2-3)—1 [(DistPT-MidPT)—ProxL]

The width/length ratio of the ectal facet of NMB Eh 719 (55) is more similar to those of adapines than notharctines (means of which, reported by Gebo et al. (2001), range from 45–52, but with ranges extending to 60 in Notharctus pugnax). The value of 64 reported for NMMP 58 would be particularly high for an adapiform, with similar values only seen in Adapis. Based on the values reported by Thalmann (1994), the index for Europolemur klatti is particularly low (41.8), at the low end of the notharctine range.

The cuboid facets of Caenopithecus are similar to those of adapines in being fan-shaped, but the long axes of the facets in the former are more dorsoplantarly oriented than the latter (Fig. 12). A similarly dorsoplantar orientation of the cuboid facet appears to be present in NMMP 20 (Ciochon et al., 2001) and NMMP 58 (Beard et al., 2007), and, among crown strepsirrhines, in lorisids and Babakotia (Fig. 13). Notharctines, asiadapines, and Anchomomys have cuboid facets whose long axes are oriented more mediolaterally than dorsoplantarly.

Figure 13 Calcanei of extant and subfossil strepsirrhines compared to NMB Eh 719, attributed here to Caenopithecus lemuroides.

(A) Varecia variegata (AMNH 201384); (B) Daubentonia madagascariensis (AMNH 185643); (C) Perodicticus potto (AMNH 184597); (D) Nycticebus coucang (AMNH 102027); (E) Babakotia radofilai (DPC 11818); (F) Caenopithecus lemuroides (NMB Eh 719). Views in, from left to right, medial, dorsal, lateral, and plantar; and, on the far right, proximal (above) and distal (below). Scale bars are equal to 5 mm.

The well-developed distal calcaneal tubercles of Caenopithecus (particularly notable in NMB Eh 719 and NMB En.269) are similar to those of adapines, which typically (but not consistently) have better-developed tubercles than notharctines such as Cantius and Notharctus. Anchomomys, Marcgodinotius, and NMMP 58 also have relatively small distal calcaneal tubercles.

The dorsolateral inclination of the sustentacular facet in Caenopithecus is—when judged relative to the mediolateral plane of the ectal facet—also seen in basal adapiforms such as Cantius and Marcgodinotius; other adapiforms, including adapines and more basal taxa such as Asiadapis, show a larger angle between the mediolateral planes of the ectal and sustentacular facets.

The strong medial bowing of the calcaneal tuber in Caenopithecus, combined with its lack of development of any lateral flaring, appears to be unique among adapiforms, but among extant primates is seen in lorisids (Fig. 13). The preserved portion of the calcaneal tuber of NMMP 58 appears to be bowed medially, but the most proximal portion of the tuber is damaged.

Automated geometric analysis of primate calcanei

The multidimensional scaling (MDS) plot from the auto3dgm analysis (Fig. 14; see Datasets S3 and S4 for 3D coordinates and MDS coordinates, respectively) appears to capture a mix of phylogenetic and functional signals. The first axis is probably related in large part to major differences in distal calcaneal elongation, with the relatively elongate calcanei of tarsiers, galagids, Microcebus, and omomyiforms having strongly positive scores, and the relatively foreshortened calcanei of non-primate euarchontans, hominoids, and cercopithecoids having strongly negative scores. Several basal extinct primate clades, such as Notharctinae, Adapinae, Asiadapinae, and Eosimiidae occupy a central zone along the first axis, presumably reflecting in part their intermediate levels of distal calcaneal elongation. Caenopithecus (NMB Eh 719) falls into a unique part of the primate calcaneal morphospace, showing no overlap with any other living or extinct taxon, but falling about mid-way between the polygons encompassing Dermoptera, Lorisidae, and Adapinae; Babakotia also falls into this part of the MDS plot, but is in an even more extreme position, actually falling outside of the morphospace encompassing all other known living and extinct primates. Non-cheirogaleid, non-lepilemurid lemuriforms show considerable cohesion along the first axis despite variation in locomotor style, presumably reflecting strong phylogenetic signal in calcaneal morphology; the same can be said of platyrrhines as a whole. Lorisids fall within the lemuriform morphospace along the first axis, but, along with Babakotia, some adapines, and Caenopithecus, have strongly positive scores along the second axis. Overall, as with the multivariate analysis of the astragalus, Caenopithecus appears to occupy a calcaneal morphospace that is best interpreted as being related to cautious slow climbing, considerable pedal mobility, and possibly hind limb suspension. The proximity of Caenopithecus to adapines is consistent with its phylogenetic placement (see below), while its more positive score along the second axis (relative to most adapines) might reflect an increased emphasis on slow and cautious climbing, and possibly hind limb suspension, when compared to its adapine relatives. Such a scenario is also consistent with the functional interpretation that has been put forth for the closely related caenopithecine Afradapis, and the possible caenopithecine Adapoides, on the basis of their astragalar morphology, which is very loris-like (Boyer, Seiffert & Simons, 2010). We further explore the functional implications of Caenopithecus’ tarsal morphology below.

Figure 14 Multidimensional scaling plot derived from automated geometric morphometric analysis of euarchontan calcanei, based on 1,200 points.

The position of Caenopithecus lemuroides (based on NMB Eh 719) is marked by a white star enclosed in a red circle. Abbreviations: “Asiad,” Asiadapinae; “Daub,” Daubentonia; “Noth,” Notharctinae; “Proteo,” Proteopithecus.

Phylogenetic analysis

Parsimony analysis of the character matrix with 256 of 291 characters ordered, transitions between fixed and polymorphic states weighted as a half-step, premolar re-acquisition not allowed, and with the molecular scaffold enforced recovered two trees of 4330.5 steps (consistency index excluding uninformative characters (CI) = 0.1605, retention index (RI) = 0.5646, rescaled consistency index (RC) = 0.0914, Fig. 15). The only differences between the current result and that of Boyer et al. (2015b, who provided the most recent modification of the matrix used here) are (1) the placement of the clade (Europolemur dunaifi, (Aframonius, (Afradapis, Caenopithecus))) as the sister taxon of Adapinae to the exclusion of Microadapis (and all other adapiforms), (2) the placement of taxa previously situated as basal caenopithecines (Europolemur klatti and Mahgarita stevensi) as members of a (Europolemur klatti, (Darwinius, (Mahgarita, Mescalerolemur))) clade, and (3) notharctines (represented by Cantius) and the “cercamoniines” Pronycticebus and Protoadapis are consecutive sister taxa of a clade containing all of the aforementioned taxa. None of these relationships are well-supported—only the Adapinae (bootstrap value of 93), Adapis parisiensis + Leptadapis magnus (91), and Afradapis + Caenopithecus (70) clades are supported by bootstrap values higher than 50.

Figure 15 Phylogenetic analysis with some multistate characters ordered.

Strict consensus of two equally parsimonious trees of length 4330.5 recovered following parsimony analysis (10,000 heuristic search replicates) of the 291 character matrix in PAUP* 4.10b, with 256 of the characters ordered and transitions between “fixed” and “polymorphic” states in ordered morphoclines weighted as 0.5. See text for tree statistics; numbers above or below branches are bootstrap values, following 1,000 pseudoreplicates. Relationships among extant species were constrained by a molecular “scaffold” following Springer et al. (2012).

Parsimony analysis with all characters unordered and equally weighted, but constrained by the molecular scaffold, led to the recovery of 342 equally parsimonious trees of length 4,638 (CI = 0.2218, RI = 0.4950, RCI = 0.1104) (Fig. 16A). In contrast to the results from the analysis with some characters ordered and scaled, the sister taxon of Adapinae was found to be a (Microadapis, (Pronycticebus + Protoadapis)) clade rather than Caenopithecinae; an Afradapis-Caenopithecus clade was placed as a more basal sister group of that clade, alongside Aframonius and E. dunaifi, followed more distantly by Magharita. Darwinius, Djebelemur, E. klatti, Mescalerolemur, a European anchomomyin clade, and an African “Anchomomys” milleri + Azibiidae clade were all placed outside of this clade, but higher up the strepsirrhine stem lineage than notharctines, asiadapines, and sivaladapids.

Figure 16 Phylogenetic analysis with all characters unordered.

(A) Strict consensus of 342 equally parsimonious trees of length 4,638 following parsimony analysis (10,000 heuristic search replicates) of the 391 character matrix in PAUP* 4.10b, with all characters unordered and equally weighted. See text for tree statistics; numbers above or below branches are bootstrap values, following 1,000 pseudoreplicates. Relationships among extant species were constrained by a molecular “scaffold” following Springer et al. (2012). Note that, unlike the consensus tree with some characters ordered and scaled, adapines form a clade with Microadapis, Pronycticebus, and Protoadapis rather than with any caenopithecine, though Caenopithecus still forms a clade with Afradapis to the exclusion of all other species. (B) “Halfcompat” (majority-rule) consensus tree following 50 million MCMC generations in MrBayes (first 25% discarded as “burn-in”). Numbers above or below branches are posterior probabilities. Relationships among extant species were constrained by a molecular “scaffold” following Springer et al. (2012). Note that caenopithecines are paraphyletic with respect to adapines given this topology, unambiguously implying re-acquisition of the upper and lower first premolar in the latter clade.

Both Bayesian phylogenetic analyses presented here reached convergence, judging from very low (<0.01) average standard deviations of split frequencies in the last generations sampled. The “halfcompat” consensus derived from Bayesian analysis with all characters unordered (Fig. 16B) provides more resolution than the comparable parsimony analysis, but few higher-level relationships among adapiforms are well-supported aside from Adapinae (posterior probability (PP = 100)), Afradapis + Caenopithecus (97), Pronycticebus + Protoadapis (99), Sivaladapidae (100), and Asiadapinae (100). This analysis differs from both parsimony analyses in placing an Afradapis-Caenopithecus clade as the sister group of Adapinae to the exclusion of Aframonius, though with very weak support. Darwinius, E. dunaifi, and Mahgarita are placed in an unresolved polytomy outside of that clade, followed more distantly by E. klatti. One of the more notable implications of this topology is that, given parsimony optimization, it unequivocally requires re-acquisition of the upper and lower first premolar, and re-evolution of a double-rooted upper and lower second premolar (i.e., from a single-rooted condition), along the stem leading to Adapinae. Also notable in this context is the placement of the possible caenopithecine Mescalerolemur as a sister taxon of Anchomomyini, far from Mahgarita. We suspect that this result is incorrect, given that Mescalerolemur and Mahgarita are such geographically and temporally unique records in the primate fossil record (and share a number of striking morphological apomorphies), but it is nevertheless interesting that the older and arguably more primitive Mescalerolemur is placed closer to crown strepsirrhines than caenopithecines, adapines, and non-anchomomyin “cercamoniines”, similar to the results recovered by Kirk & Williams (2011) for Mescalerolemur + Mahgarita. Outside of Strepsirrhini, is also notable that, within Haplorhini, results are radically different from those in the parsimony analyses in supporting a “strict tarsier-anthropoid clade” (i.e., tarsiers join anthropoids to the exclusion of all omomyiforms) rather than a monophyletic Tarsiiformes (tarsiers + omomyiforms).

Parsimony and Bayesian analysis of the matrix with standard polymorphic scoring (which reduced the total number of states for each of the 256 multistate characters that were ordered in the parsimony analysis, thereby allowing them to be treated as ordered in MrBayes) resulted in some differences from the parsimony results in Figs. 15 and 16A, and the Bayesian results in Fig. 16B, but none that impacted the placement of Caenopithecus close to Afradapis, and, more distantly, Adapinae (Figs. 17A and 17B). In the parsimony analysis with standard polymorphic scoring (783 equally parsimonious trees of length 3796; CI excluding uninformative characters = 0.1771, RI = 0.5919, RCI = 0.1059), the only changes among stem strepsirrhines were outside of Adapidae, specifically (1) the joining of Darwinius and Europolemur klatti as a sister clade of Mahgarita + Mescalerolemur (rather than being paraphyletic with respect to Mahgarita + Mescalerolemur), (2) recovery of a Pronycticebus + Protoadapis clade as sister of the clade containing Adapidae, Darwinius, E. klatti, Mahgarita, and Mescalerolemur (rather than being paraphyletic with respect to that clade), (3) recovery of a clade containing European Anchomomys species and movement of African “Anchomomys” milleri to a sister taxon relationship with Djebelemur, and (4) placement of Mazateronodon as the sister taxon of a ((Plesiopithecus, (“A.” milleri, Djebelemur)), crown Strepsirrhini) clade.

Figure 17 Phylogenetic analysis with standard polymorphic scoring and 256 multistate characters treated as ordered.

(A) Strict consensus of 783 equally parsimonious trees of length 3,796 following parsimony analysis (10,000 heuristic search replicates) of the 291 character matrix in PAUP* 4.10b, with all characters equally weighted, “standard” scoring of polymorphisms, and 256 multistate characters treated as ordered. See text for tree statistics; numbers above or below branches are bootstrap values, following 1,000 pseudoreplicates. Relationships among extant species were constrained by a molecular “scaffold” following Springer et al. (2012). (B) “Halfcompat” (majority-rule) consensus tree following 50 million MCMC generations of the same matrix in MrBayes (first 25% discarded as “burn-in”). Numbers above or below branches are posterior probabilities. Relationships among extant species were constrained by a molecular “scaffold” following Springer et al. (2012). Note that, as in the Bayesian analysis of unordered characters caenopithecines are paraphyletic with respect to adapines given this topology, unambiguously implying re-acquisition of the upper and lower first premolar in the latter clade.

The Bayesian analysis with 256 characters ordered showed more differences from the results based entirely on unordered characters, but again Afradapis was placed as the sister taxon of Caenopithecus, and, with Aframonius, all caenopithecines were placed as sister taxa of Adapinae, but with higher support (posterior probability of 79 versus 66). There was also moderate support (PP = 72) for placement of E. dunaifi as the sister taxon of that clade. There was no support for a clade containing all stem and crown strepsirrhines to the exclusion of asiadapines, notharctines, and Donrussellia (supported by a PP of 90 in the analysis with all characters unordered); instead sivaladapids, asiadapines, and a (Microadapis (Pronycticebus, Protoadapis)) clade were placed in a basal polytomy, with only notharctines and Donrussellia being placed more basally. The anchomomyin clade broke down into a polytomy with azibiids, and Mescalerolemur was again separated from Mahgarita and placed as a sister of a clade containing anchomomyins, azibiids, djebelemurids, Plesiopithecus, and crown strepsirrhines.

When the same phylogenetic analyses are run with the Egerkingen tarsals scored as Leptadapis priscus rather than as Caenopithecus, the results of some analyses were different, though not radically so. Under parsimony (with 256 characters ordered, transitions between fixed and polymorphic states weighted as a half-step, premolar re-acquisition not allowed, and with the molecular scaffold enforced) Caenopithecus still formed a clade with Afradapis, but caenopithecines as a whole were not placed as the closest sister taxa of Adapinae; rather, the sister group of the “core” caenopithecines (i.e., (Aframonius, (Afradapis, Caenopithecus))) was a clade containing Darwinius and E. dunaifi. Mescalerolemur and Mahgarita together formed the sister group of that larger clade, and all of those taxa were joined more basally by E. klatti, while Microadapis was placed as the sister taxon of Adapinae. When all characters were treated as unordered, parsimony analysis placed adapines as the sister taxon of a (Microadapis, (Pronycticebus, Protoadapis)) clade, and caenopithecines (broadly defined) were paraphyletic with respect to that larger clade. Under Bayesian inference, the support for the paraphyly of caenopithecines with respect to adapines disappeared, and there was only very weak support (0.51 posterior probability) for an (Aframonius, (Afradapis, Caenopithecus)) clade, and no support (i.e., posterior probability of <50) for a caenopithecine-adapine clade. With the change to the treatment of polymorphic characters (i.e., use of “standard” polymorphic scoring), the same strict consensus and “halfcompat” topologies were retrieved regardless of whether the Egerkingen tarsals were attributed to L. priscus or Caenopithecus.

Discussion and Conclusions

Functional considerations

In her doctoral dissertation, Dagosto argued that “Adapis parisiensis, Leptadapis magnus, and Caenopithecus lemuroides have features of the upper ankle joint and foot proportions which strongly suggest that quadrupedal slow climbing was the dominant form of locomotion” (Dagosto, 1986, p. 333). Our analyses lend additional support to this hypothesis, and our functional interpretation is broadly consistent with that of Dagosto (1986).

It has been argued that several features of the strepsirrhine hind limb, including astragalar features such as laterally flaring fibular facets and laterally placed grooves for the tendon of flexor fibularis, reflect an ancestral dependence on the use of inverted and abducted foot postures on inclined and vertical small-diameter supports (Gebo, 2011). Some features of the Caenopithecus tarsals suggest that, relative to other adapiforms, C. lemuroides’ tarsus may have been held in particularly inverted postures. For instance, if the fibular facet angle faithfully reflects pedal inversion (Gebo, 2011; Boyer & Seiffert, 2013), then Caenopithecus’ high value alone would suggest that this taxon might have had more inverted foot postures than any other Paleogene primate aside from Afradapis, Cantius nuniensis, and Djebelemur (though the ranges of Adapis parisiensis and Notharctus tenebrosus also encompass values as high as that of Caenopithecus, see Boyer & Seiffert (2013)). Boyer et al. (2015a) and Boyer et al. (2015b) have also recently argued that the ratio of medial tibial facet (MTF) area to ectal facet (EF) area provides another quantitative proxy for pedal inversion in fossil primates, and Caenopithecus’ ratio of 0.11 is higher than that of any other adapiform. Among extant primates, Caenopithecus’ value is higher than those of cheirogaleids, Lepilemur, Daubentonia, and even some lorises, but is equaled or exceeded by leaping galagids, indriids, and lemurids. Boyer et al. (2015b) argued that the high MTF/EF area ratios of the latter strepsirrhines might reflect increased loading of the MTF due to acrobatic grasp-leaping, i.e., increased loading relative to more generalized (less acrobatic) ancestors whose feet were already more habitually inverted than those of other, non-strepsirrhine primates, and thus already had high MTF/EF area ratios.

Given that extant strepsirrhine grasp-leapers have both high fibular facet angles and high MTF/EF area ratios as in Caenopithecus, it could be argued that the latter’s values reflect acrobatic leaping as well as pedal inversion. However, we consider this to be less likely than a dependence on slow and cautious climbing, and possibly hind limb suspension, when other loris-like features of Caenopithecus’ tarsals are taken into account, such as its high astragalar neck angle, inferred presence (based on neck width) of a broad astragalar head, a curved distal projection of the medial tibial facet that extends onto the astragalar neck, and a deeply excavated groove for the tendon of flexor fibularis on the plantar surface of the body (which, at least among extant strepsirrhines, is deepest in species that are known to habitually engage in hind limb suspension). Calcaneal features seen in Caenopithecus that are likely related to increased pedal inversion and overall mobility rather than acrobatic grasp-leaping include the dorsoplantarly oriented long axis of the cuboid facet, projection of the ectal facet dorsal to the calcaneal tuber (rather than being level to it), the dorsolateral inclination of the sustentacular shelf, and the convexity of the lateral border of the calcaneal body and (correlated) medial bowing of the calcaneal tuber. Furthermore, the distal calcaneus of Caenopithecus is less elongate (relative to body mass) than are the calcanei of acrobatic lemuriforms with high MTF/EF ratios. Finally, in our morphometric analyses the tarsals of Caenopithecus show no close phenetic affinities to those of extant leaping primates, and instead appear to be most consistent with a somewhat loris-like complex derived from a generalized adapiform Bauplan that combined both notharctine-like and adapine-like features.

The trochlea-like groove for the tendon of flexor fibularis (the medial wall of which is highly abraded on NMB En.270, but obviously quite prominent, see Fig. 18, feature 2) presumably served to prevent medial slippage of that tendon and maintain its line of action while the foot was held in diverse inverted postures, including those that would have been required by hind limb suspension. The strong medial buttressing of this groove via a bony plantar projection might have been particularly important for Caenopithecus because, unlike most other primates, the plantar groove guiding the passage of the flexor fibularis tendon along the plantar surface of the calcaneal sustentaculum was poorly developed, as might be expected given the dorsolateral inclination of the sustentacular shelf. The plantar projection on the medial aspect of the astragalar body might, then, represent a compensatory feature that maintained a tunnel-like passage for the tendon. The continuation of this trochlear passage for the flexor fibularis tendon onto the plantar surface of the astragalus could have been particularly important for the maintenance of strong pedal grasps when the foot was held in extreme plantarflexion (as would occur during hind limb suspension), as this groove and its associated ligaments would have formed—and maintained the integrity of—the so-called “tarsal tunnel” (e.g., Keck, 1962), a structure that, in the case of suspensory species that habitually hold the foot in extreme plantarflexion, might also prevent plantar bowstringing of the flexor fibularis tendon. This inference is supported by the presence of a similarly deep plantar exposure of the flexor fibularis groove in occasionally or habitually suspensory euarchontans such as Cynocephalus, subfossil palaeopropithecid “sloth lemurs,” lorises, and even in the occasionally suspensory lemurid Varecia variegata (Meldrum, Dagosto & White, 1997), a species whose flexor fibularis grooves are comparatively more shallow than the former taxa, but are nevertheless larger and deeper than those of its close lemurid relatives that rarely, if ever, engage in hind limb suspension (Fig. 11). All of these species also resemble Caenopithecus in lacking the distinct posterior trochlear shelf that is seen various Paleogene primates (e.g., notharctine adapiforms and microchoerine omomyiforms) whose postcranial morphology suggests an increased capacity for leaping; the absence of such a shelf-like projection along the proximal border of the lateral tibial facet presumably would have allowed for increased plantarflexion relative to those taxa that have a shelf. A prominent medial wall of the flexor fibularis groove that is similar to that of Caenopithecus is also present in Babakotia (Fig. 9H). In extant lorises the same function is achieved in a slightly different manner; rather than having a mediolaterally restricted plantar projection, instead the lateral tibial facet (=trochlear articular surface) is mediolaterally broad and its entire surface projects plantar to the flexor fibularis groove (Figs. 9E and 9F), providing a structurally somewhat different (but functionally similar) strong medial buttress for a similarly trochlear groove—again presumably preventing medial slippage and bowstringing of the flexor fibularis tendon, and maintaining the line of action of the tendon during strong pedal grasping and hind limb suspension. Quantification of flexor fibularis groove depth demonstrates that Afradapis and Caenopithecus are both loris-like in this feature (Fig. 11), while Babakotia and Megaladapis show the same pattern but take it to an extreme, with values that are much higher than those of any other strepsirrhines. Though Varecia isn’t the only lemurid that engages in hind limb suspension (and it should be noted that occasional use of hind limb suspension is not at all uncommon among primates (e.g., Meldrum, Dagosto & White, 1997)), it does so more frequently, and its high values relative to its lemurid relatives might also be explained as a correlate of this behavioral pattern. The high value of Cheirogaleus medius is interesting given evidence from that species’ axial skeleton for a more loris-like dorso-stable Bauplan than other cheirogaleids (including Cheirogaleus major) (Granatosky et al., 2014). The high value of the terrestrial subfossil form Archaeolemur is more difficult to explain and requires further investigation, though it does not approach the magnitude seen in Afradapis and Caenopithecus. In summary, the pronounced height of the astragalar body in Caenopithecus—which is often associated with leaping propensities in primates—seems more likely to be a correlate of the plantar projection that buttresses the flexor fibularis groove, while at the same time being accentuated by the absence of the posterior shelf. We interpret these correlated features as two aspects of a morphological pattern that more likely facilitated inversion, increased capacity for plantarflexion, and possibly hind limb suspension in this species.

Figure 18 Sustentacular facet morphology and flexor fibularis groove depth in Caenopithecus and other adapiforms.

Stereopairs of (A) Caenopithecus (NMB En.270, reversed); (B) Afradapis (DPC 21445C, reversed); (C) Adapis (ECA 7377); and Asiadapis (GU 747) in distal view, showing (1) the laterally expanded convexity of the sustentacular facet in Afradapis and particularly Caenopithecus, and (2) the deep flexor fibularis grooves of Afradapis and Caenopithecus when compared with Adapis and Asiadapis.

The dorsoplantar orientation of the long axis of the cuboid facet (and correlated medial placement of the cuboid pivot) is also seen in slow-moving lorises and palaeopropithecids, and suggests that the articulating cuboid would have been held in such a position that the navicular articulation was more dorsally oriented than in those taxa that have cuboid facets whose “pits” are situated along the plantar margin of the facet. The dorsolateral inclination of the calcaneal sustentacular facet might represent a structural mechanism that would have restricted movement of the astragalar neck such that, in the most stable positions allowed by the articulating ectal and sustentacular facets, the long (roughly mediolateral) axis of the astragalar neck and head would have been held in an inverted position. A similar dorsolateral orientation of the sustentaculum, with no plantar groove, is also seen in Babakotia (Fig. 13E). The tightly curved calcaneal ectal facet of Caenopithecus nevertheless appears to provide an extensive surface for articulation of the astragalar ectal facet, thereby allowing for considerable mobility in the proximodistal plane and corresponding potential for inversion (when positioned relatively proximally) and eversion (when positioned relatively distally) of the calcaneus with respect to the astragalus (Fig. 19, and see Decker & Szalay, 1974) (though this assessment is based on articulation with NMB En.270, which probably does not belong to the same individual, see Fig. 19). This capacity for eversion is interesting given the strikingly convex shape of the proximal portion of the sustentacular facet of Caenopithecus (Fig. 16A), which is fairly unique among adapiforms (see the relatively flat sustentacular facets of Adapis and Asiadapis, Figs. 18C and 18D) and rare among extant primates. Given that the calcaneal sustentacular facet is relatively flat (mediolaterally), the convexity of the articulating astragalar sustentacular facet, and the medioplantar orientation of the medial extension of that facet, suggests a mechanism by which the astragalus might be capable of slipping into a more or less “locked” (but not particularly stable) position when the calcaneus is everted relative to the astragalus (Figs. 19B, 19D, 19E and 19H). Such everted positions greatly reduce the overlap of the articulating sustentacular facets, however, and furthermore lead to a corresponding separation of the head of the astragalus from the distal calcaneus (and, presumably, correlated separation of the navicular from the cuboid), suggesting to us that these foot postures are unlikely to have been used habitually. It is conceivable that such postures might have been employed during quadrumanous bridging behaviors, for instance when one foot maintains a strong grasp on an inclined support, but becomes increasingly everted as the body turns away from that support to grasp a nearby branch with the other limbs. A similar medioplantar extension of the astragalar sustentacular facet is also seen in lorisids, lending some support to that interpretation, but in these taxa the facets are not nearly as convex as that of Caenopithecus.

Figure 19 Articulation of the unassociated Caenopithecus astragalus NMB En.270 and and calcaneus NMB Eh 719 in inversion and eversion.

Articulated astragalus and calcaneus in (A) lateral view, inverted; (B) lateral view, everted; (C) medial view, inverted; (D) medial view, everted; (E) dorsal view, inverted; (F) dorsal view, everted; (G) distal view, inverted; (H) distal view, everted. In each unique view, from top to bottom the astragalus is rendered as solid (e.g., A1), translucent (e.g., A2), or is not shown (e.g., A3).

Among living primates, projection of the ectal facet dorsal to the calcaneal tuber is seen in generally slow-moving species such as lorisids, whereas leaping species that depend on powerful propulsion via the triceps surae musculature (such as tarsiers; Gebo, 1987), tend to have calcaneal tubers that project dorsally to approximately the same level as the most dorsal surface of the ectal facet. The development of rugosities along the medial margin of the calcaneal tuberosity on NMB Eh 719 suggests that the calcaneal tendon might have had strong insertions along both the dorsal and medial surfaces of the tuber, as might be required if the calcaneus as a whole was typically held in an inverted position with respect to the tibia. The medial curvature, and moderately developed plantar projection, of the calcaneal tuber would also be expected to have increased the mechanical advantage of the superficial head of the flexor digitorum brevis muscle, which is important for species that engage in hind limb suspension because such an arrangement allows for strong digital flexion in diverse foot postures, notably those that might reduce the contractile potential of flexor fibularis (Sarmiento, 1983). A distinctly convex lateral border of the calcaneus is also seen (though to a greater degree) in lorisids, whereas dedicated leapers tend to have calcaneal tubers and distal segments whose long axes are roughly parallel. The mediolateral curvature of the calcaneal body in primates with more inverted feet might be functionally similar to dorsoplantar curvature of the calcaneal body in primates that have more everted feet; i.e., due to inversion, the medial surface of the tuber becomes functionally equivalent (in terms of the line of action of the triceps surae musculature) to the dorsal surface of the tuber in taxa with more everted feet.

Finally, we note that Caenopithecus has a particularly well-developed tubercle on the lateral surface of the astragalar neck for the astragalar-ectocuneiform ligament. This ligament tubercle is also well-developed in lorises, highly suspensory sloth lemurs such as Babakotia and Palaeopropithecus, the loris-like caenopithecine Afradapis, and the loris-like possible caenopithecine Adapoides, but is not clearly expressed to the same extent in adapines or grasp-leaping lemuriforms. We infer that the large tubercle reflects the existence of a particularly large astragalar-ectocuneiform ligament, and hypothesize that such an enlarged ligament might have been one of several ligamentous features that served (at least in part) to keep the astragalus closely anchored to the surrounding tarsals when the compressive load of the tibia on the astragalus was released—as would be expected to occur during hind limb suspension. Such a morphological adaptation would undoutedly be a more energy-efficient mechanism for maintaining articular integrity of the tarsals in habitually suspensory postures than to depend on muscles and their associated tendons. This interpretation calls into question whether Notharctus might have also utilized hind limb suspensory behaviors to a significant degree because it (but not Cantius) exhibits a tubercle that is developed to the same extent as that of Caenopithecus.

In summary, when all of the evidence presented here is taken into account, we consider it most likely that Caenopithecus had a tarsus that was habitually held in strong inversion, and that this species was capable of strong pedal grasping in a diversity of postures, as is typical of slow-moving primates that simultaneously use all four limbs to navigate arboreal settings in which branches are small relative to hand and foot size (such as terminal branches). We do not detect any compelling morphological evidence for habitual leaping or acrobatic grasp-leaping in Caenopithecus’ tarsals, but we do consider their morphology to be consistent with the use of hind limb suspension. Jenkins & McClearn (1984) pointed out that hind limb reversal of the sort that is required for hind limb suspension is accomplished in placental mammals via cruro-astragalar plantarflexion, subtalar inversion, and transverse tarsal supination, all of which we consider to have been possible in Caenopithecus and facilitated particularly well by its morphology. Inverted postures might be further indicated by a long and deep plantar exposure of the flexor fibularis groove, the development of the large tubercle for the astragalar-ectocuneiform ligament, and by the correlation of such extreme postures with slow climbing as suggested by the existence in Caenopithecus of relatively small tarsal facets (in this case estimated relative to tooth size—though admittedly this could also relate to a folivorous diet (and relatively large molars) as suggested above).

Lifestyle, phylogenetic position, and biogeographic origin of Caenopithecus

If we are correct in attributing the Egerkingen tarsals to Caenopithecus lemuroides, then members of this species can now be reconstructed as having been folivorous slow climbers that were approximately 1.5–2.5 kg in body mass. We consider it unlikely that C. lemuroides individuals were adept leapers, and more probable that members of this species consistently maintained powerful grasps on branches as they moved through their arboreal habitats. We also consider it possible, based in part on the form-function correlation that we propose for the depth of the flexor fibularis groove on the plantar surface of the astragalus, that C. lemuroides might have regularly engaged in hind limb suspensory postures, though more evidence from the other bones of the hind limb is certainly needed to fully test that hypothesis. Regardless of whether the inference of hind limb suspension is correct, the reconstruction of C. lemuroides as a folivorous and slow climbing species suggests that the best extant primate analogues might be found among howler monkeys (Alouatta), although all known living and extinct alouattines are considerably larger than C. lemuroides (ranging in size from ∼4 to 11.5 kg; Smith & Jungers, 1997). The combination of folivory and slow climbing arguably also fits with what might be predicted for the adaptive profile of basal “stem” members of the lemuriform indrioid palaeopropithecid clade, prior to their acquisition of extreme specializations for quadrumanual under-branch suspension. The acquisition of such a lifestyle in distantly related lineages that presumably shared a common ancestor with distinctively strepsirrhine ankle morphology (Gebo, 2011; Boyer & Seiffert, 2013) may help to explain tarsal similarities that are shared by C. lemuroides and Babakotia.

Most of our phylogenetic analyses place Caenopithecus close to adapines to the exclusion of other non-caenopithecine adapiforms, but importantly the new tarsal evidence shows that Caenopithecus is not just a dentally aberrant adapid with an adapine-like postcranium; instead, Caenopithecus appears to be an adaptively unique member of Europe’s middle Eocene primate fauna. The biogeographic underpinnings of the adapiform diversity seen in the middle Eocene of Europe remain mysterious. Godinot (1998) argued that Caenopithecus and Adapinae were likely immigrant taxa that arrived in Europe independently during the middle Eocene, possibly from Asia, but simple parsimony reconstructions of continental biogeography onto the trees derived from our phylogenetic analyses all unambiguously imply that the last common ancestor of caenopithecines and adapines was already present in Europe. Nevertheless, a key taxon in Godinot’s scenario was middle Eocene Adapoides from the Shanghuang fissure fillings in China (Beard et al., 1994), which was not included in our phylogenetic analyses because undescribed material of that taxon announced in an abstract (Coster, Ni & Beard, 2012) indicates that teeth previously assigned to another adapiform in the Shanghuang fauna actually belong to Adapoides; we await explicit clarification on this matter before integrating Adapoides into phylogenetic analysis. Admittedly, the new observations (and expanded sample of Adapoides noted by Coster et al.) could have an important impact on our understanding of the biogeographic origins of adapines and caenopithecines.

The “core” caenopithecines (Aframonius, Afradapis, and Caenopithecus) present another biogeographic puzzle—depending on the assumptions and methods employed there were either (1) two unambiguously independent caenopithecine dispersals from Europe to account for the presence of Aframonius and Afradapis in Africa (i.e., on the two “allcompat” trees derived from the Bayesian analyses, and on that derived from parsimony analysis with all characters unordered), or (2) ambiguity, with two independent dispersals to Africa or back-migration of Caenopithecus to Europe being equally parsimonious (i.e., on the remaining trees). The biogeographic scenario is further complicated by the fact that the middle Eocene “caenopithecids” Mescalerolemur and Mahgarita from west Texas—which are unique records in North America, and arguably the most biogeographically anomalous adapiforms—do not form a clade in our Bayesian analyses, or in our parsimony analyses when characters are treated as unordered (these taxa do form a clade in our parsimony analyses, when some characters are treated as ordered). In our opinion, the separation of these genera (and the independent colonizations of North America required by such a result) is almost certainly erroneus and more likely reflects the in situ acquisition of caenopithecine-like convergences in Mahgarita (e.g., mandibular symphyseal fusion, enlarged upper molar hypocones, acquisition of an enlarged P3 protocone) from a Mescalerolemur-like form that more closely resembled anchomomyins or djebelemurids. The placement of Mescalerolemur as a close relative of crown strepsirrhines to the exclusion of non-anchomomyin adapiforms in our Bayesian analyses is similar to the placement retrieved by Kirk & Williams (2011) for a combined Mescalerolemur-Mahgarita clade when those authors employed a previous version of the matrix used here in their parsimony analyses (but notably with a reduced taxon set outside of Strepsirrhini). The divergent placements of Mescalerolemur and Mahgarita in some of our phylogenetic analyses clearly reveals the potential for parallel acquisition of the aforementioned caenopithecine-like features from more generalized small-bodied ancestors, and suggests that much more fossil evidence is needed from Afro-Arabia, Asia, Europe, and North America to tease apart the phylogenetic and biogeographic history of this group. Tarsals of Mescalerolemur or Mahgarita could provide particularly decisive evidence, given that Adapoides, Afradapis, and Caenopithecus are all known to have somewhat loris-like astragali (Gebo et al., 2001; Boyer, Seiffert & Simons, 2010) that differ markedly from those known for notharctids, djebelemurids, anchomomyins, and Europolemur.

Finally, we note that middle Eocene localities in Europe have now yielded primate and non-primate species whose adaptive profiles closely resemble those that might have been present among Madagascar’s lemuriforms in the later Oligocene and early Miocene, when non-daubentoniid clades were likely diversifying (Springer et al., 2012; Kistler et al., 2015). As discussed earlier, as slow moving folivores, caenopithecines might have resembled basal palaeopropithecids, while adapines and Europolemur-like forms were presumably more like basal lemurids in their lifestyles; anchomomyin stem strepsirrhines and microchoerine omomyiforms were cheirogaleid-like in being small-bodied insectivorous or omnivorous species capable of leaping (Dagosto & Schmid, 1996; Boyer et al., 2015b); and the non-primate apatemyids appear to have occupied a niche similar to that of extant Daubentonia (e.g., Koenigswald, 1990). A similar pattern might have existed slightly later in the middle Paleogene of Afro-Arabia, with the caenopithecine Afradapis occupying a basal palaeopropithecid-like niche, djebelemurids and basal lorisiforms occupying cheirogaleid-like niches, and plesiopithecids (based solely on craniodental evidence) possibly being somewhat Daubentonia-like (Godinot, 2006; Godinot, 2010). Afro-Arabia clearly differs from both Europe and (presumably) Madagascar, however, in having a diverse anthropoid fauna through the later Paleogene. The extent to which dispersal and/or endemic common ancestry shaped the similar adaptive composition of primate communities on these landmasses (which were largely isolated and thus biogeographically filtered during the later Paleogene) is a complex puzzle that will provide fascinating insight on primate evolutionary history as it is gradually pieced together by future work.

Supplemental Information

Appendix S1 Measurements on tarsal facets and lower second molars of extant and subfossil euarchontans that were used in regression analyses

Click here for additional data file.

Appendix S2 Measurements on tarsal facets and lower second molars of Paleogene euarchontans that were used in regression analyses

Click here for additional data file.

Appendix S3 Raw measurements and angular measurements, originally published by Boyer, Seiffert & Simons (2010) and augmented by Chester et al. (2015), taken on living and extinct euarchontans that were converted to shape variables and radians for principal components analysis

Asterisks indicate which of the measurements were estimated on NMB En.270 due to abrasion or damage. More detailed explanations of how measurements are taken on digital models can be found in Boyer, Seiffert & Simons (2010).

Click here for additional data file.

Appendix S4 Data from dental topography analysis of Caenopithecus, Afradapis, and 109 individuals from 21 extant “prosimian” genera

Values are provided for the input variables M2 Relief Index (RFI), M2 Orientation Patch Count (OPC), and M1 area (length x width; note that the same M1 area (i.e., the mean M1 area of all M1s associated with M2s) is used for each M2 of Afradapis and Caenopithecus, because associated M1s were not available for each M2 analyzed); the dietary classification for extant taxa (“Actual group”); the group predicted by the discriminant function analysis (“Predicted group (highest prob.)”); probability of classification into the group predicted by the discriminant function analysis (“Probability”); next-best classification predicted by the discriminant function analysis (“Second highest group”); probability of the next-best classification predicted by the discriminant function analysis (“Probability (Second highest group)”); and the discriminant function scores for the first three discriminant axes (“DF1”, “DF2”, “DF3”). Results are based on a stepwise discriminant function analysis (using the Mahalanobis distance method) run in SPSS v. 22; because Box’s M test was significant, classifications were based on the group covariance matrices of the canonical discriminant functions, not the original variables. 93.6% of the original grouped cases were classified correctly. See Winchester et al. (2014) for a recent summary of the RFI and OPC methods and their efficacy.

Click here for additional data file.

Figure S1 Plot of the first two discriminant functions based on RFI, OPC, and M1 area

Individuals from each dietary category are enclosed by ellipses that include 95% confidence intervals (calculated in PAST, (Hammer, Harper & Ryan, 2001). Note that two specimens of Afradapis had the same RFI and OPC values and plot at the same points along DF1 and DF2.

Click here for additional data file.

Dataset S1 Character-taxon matrix employed in the parsimony analysis with some multistate characters ordered and scaled

Transitions between “fixed” states are equal to a single step (weights are provided in the PAUP block). Note that a constraint tree with the following topology must be enforced to obtain the results presented here: (Tupaia glis, (((((Saimiri sciureus, Aotus trivirgatus), Alouatta seniculus), (Allenopithecus nigroviridis, Pan troglodytes)), Tarsius bancanus), ((Propithecus spp., ((Cheirogaleus major, Microcebus murinus), Lepilemur mustelinus), (Lemur catta, Varecia variegata)), (((Loris tardigradus, Nycticebus coucang), (Arctocebus calabarensis, Perodicticus potto)), (Galagoides demidoff, (Galago moholi,Otolemur crassicaudatus)))))).

Click here for additional data file.

Dataset S2 Constraint tree used in the parsimony analyses (also enforced as partial constraints in MrBayes)

Click here for additional data file.

Dataset S3 3D coordinates for specimens analyzed in the automated geometric analysis of calcanei

Click here for additional data file.

Dataset S4 Multidimensional Scaling coordinates derived from the automated geometric analysis of calcanei

Click here for additional data file.

We thank the Academic Editor Brian Kraatz, as well as Daniel Gebo, Laurent Marivaux, and one anonymous reviewer for their helpful comments on the manuscript. ERS thanks Marcelo Sánchez-Villagra for coordinating travel to Switzerland in 2011, which ultimately led to the re-identification of the Caenopithecus tarsals in the NMB collections. We thank the following curators, collections managers, and colleagues for access to comparative material: K Beard (University of Kansas); P Chatrath, G Gunnell, E Simons (Duke University); S Chester (Hunter College); H Covert and T Culver (University of Colorado); E Delson, W Harcourt-Smith (Lehman College); N Duncan, J Galkin, J Meng, R O’Leary, I Rutzky, N Simmons, and E Westwig (American Museum of Natural History); R Dunn (Des Moines University); D Gebo (Northern Illinois University); M Godinot (Ecole Pratique des Hautes Etudes); P Holroyd (University of California Museum of Paleontology); L Marivaux (Institut des Sciences de l’Évolution de Montpellier); L Tallman (Grand Valley State University); P Gingerich (University of Michigan); K Rose (Johns Hopkins University); C Sidor (Burke Museum); T Smith (Royal Belgian Institute of Natural Sciences); and A Su (Cleveland State University). We thank J Puente for running initial versions of the auto3dgm and T Gao for helping establish a user friendly version of the MATLAB code for auto3dgm. We thank Justin Gladman for taking photographs of the tarsal material and helping with some early versions of analyses.

Institutional abbreviations

AMNH American Museum of Natural History, New York, NY, USA

CM Carnegie Museum of Natural History, Pittsburgh, Pennsylvania, USA

DPC Duke Lemur Center Division of Fossil Primates, Durham, North Carolina, USA

ESC Private collection of Mr. Dominique Vidalenc (Escamps locality)

GMH Geiseltalmuseum, Halle, Germany

GU H.N.B. Garhwal University, Srinagar, Uttarakhand, India

HTB Hamann-Todd collection, Cleveland Museum of Natural History

IPS Institut de Paleontologia de Sabadell (= Institut Català de Paleontologia Miquel Crusafont), Spain

ISEM Institut des Sciences de l’Évolution de Montpellier, Montpellier, France (ISEM-ECA, Escamps locality, ISEM-BFI, La Bouffie locality)

IRSNB Institut Royal des Sciences Naturelles del Belgique, Brussels, Belgium

IVPP Insitute of Paleontology and Paleoanthropology, Chinese Academy of Sciences, Beijing, China

Ma-PhQ Muséum d’Histoire Naturelle Victor Brun, Montauban, France

MCZ Museum of Comparative Zoology, Harvard University, Cambridge, Massachusetts, USA

MNHN Muséum National d’Histoire Naturelle, Paris, France

NMB Naturhistorisches Museum Basel, Basel, Switzerland

NMNH Smithsonian Institution National Museum of Natural History, Washington, D.C., USA

PMZ Paleontological Museum of the University of Zürich, Zürich, Switzerland

ROS Private collection of Mr. Dominique Vidalenc (Rosieres locality)

SBU Department of Anatomical Sciences, Stony Brook University, Stony Brook, New York, USA

SDNHM San Diego Natural History Museum, San Diego, California, USA

UCM University of Colorado Museum of Natural History, Boulder, Colorado, USA

UCMP University of California Museum of Paleontology, Berkeley, California, USA

UF University of Florida, Florida Museum of Natural History, Gainesville, Florida, USA

UM University of Michigan, Ann Arbor, Michigan, USA

USGS U.S. Geological Survey, Denver, Colorado, USA

UNSM University of Nebraska Science Museum, Lincoln, Nebraska, USA

USNM United States National Museum, Smithsonian Institute, Washington D.C., USA

UWBM Burke Museum, Seattle, Washington, USA

VPL/JU Vertebrate Palaeontology Laboratory, University of Jammu

YPM Yale Peabody Museum, Yale University, New Haven, Connecticut, USA

Additional Information and Declarations

Competing Interests

Author Contributions

Data Deposition

1 We were unaware of Dr. Dagosto’s earlier work at the time that this manuscript was submitted for review, and we thank Dr. DL Gebo for bringing this to our attention.

2 It should be noted that Abel’s “Tarsioidea” also included taxa that are now considered to be plesiadapiforms (e.g., Paromomys, Carpolestes) or adapiforms (Anchomomys, Periconodon, Pronycticebus). Other adapiforms (Adapis, Pelycodus, Notharctus, Protoadapis) were included in his Lemuroidea.

Loïc Costeur is an employee of Naturhistorisches Museum Basel.

Erik R. Seiffert and Doug M. Boyer conceived and designed the experiments, performed the experiments, analyzed the data, contributed reagents/materials/analysis tools, wrote the paper, prepared figures and/or tables, reviewed drafts of the paper.

Loïc Costeur reviewed drafts of the paper.

The following information was supplied regarding the deposition of related data:

Digital models of the specimens described here are available on MorphoSource (www.morphosource.org). DOIs for the specimens are as follows:

DOI 10.17602/M2/M5397

DOI 10.17602/M2/M5398

DOI 10.17602/M2/M5399

DOI 10.17602/M2/M5400

DOI 10.17602/M2/M5401

DOI 10.17602/M2/M5402

DOI 10.17602/M2/M5403

DOI 10.17602/M2/M5404.

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
