# Peer review of "Primate tarsal bones from Egerkingen, Switzerland, attributable to the middle Eocene adapiform Caenopithecus lemuroides"

_PeerJ, doi:10.7717/peerj.1036_

## Round 0.1 · original submission · Minor Revisions

This is a thorough, well figured, clearly written paper that will be an important addition to the literature. I agree with all three reviewers in that the manuscript is strong, but echo here several issues that must be clarified before publication.

Reviewer #1 points out that this material was dealt with in the dissertation of Dagasto (1986), and given that work, and in particular the care in which the authors have outlined the history of study, please update the manuscript to discuss Dagosto’s (1986) treatment of these specimens.

As noted by several reviewers, please consider what other sivaladapids have been described.

I agree with reviewer #3 that there should be acknowledgement of collections and persons used to acquire these data in the acknowledgement section.

The organization and reference to supplemental and table material needs improvement. There is some ambiguity among figures, tables, and supplemental data as to what specimens and specimen measurements were used.

It seems that tables 1 and 2 were inadvertently left out of the submission. Is App. S4 meant to be Table 2? There is no reference to app. S4 in the body of the manuscript.

S1 – Why are specimen numbers not listed for all specimens?

S2 - There are no explanations of these measurement numbers (likely referred to in Boyer [2010], and meant to be explained in the missing(?) table 2? Were any of the reported measurements estimates? If so, note which ones.

S4 – No reference in text.

As referenced by reviewer #3, there should be a clean appendix (or table), in which all the NMB specimens studied are listed, and measurements reported (included noted where measurements are estimates). This seems to be buried in various appendices. For example, where are the measurements used to ‘Cl’ and ‘Lp’ in figure 2? Please consistently reference where measurements are located in figure captions.

It’s unclear whether only NMB 719 & 270 were solely used for analyses, or if other, less well-preserved specimens were also used at times. In your figures, please note specimen numbers in plots if only one specimen was used. I think this would clarify if means were being used.

Figure 2: Where are measurements reported?
Figure 3: Where are measurements reported?
Figure 10: please reference S2, but also, what measurements are estimates?
Figure 11: No reference to specimen numbers in figure caption (or where measurements are located)

I believe the Smithsonian has updated their acronym to USNM, as the authors correctly use.

What software was used for statistical analyses, how were measurements taken?

More broadly, two general suggestions for the authors:

I wonder if the description and qualitative comparison would fit better before the quantitative results are given, with figures shifted accordingly. The ‘functional considerations’ seem to fit better in the discussions section.

Be sure to respond, specifically, to all other reviewer comments.

·

Basic reporting

This a well written and well thought out analysis of several foot bone elements from a site (Egerkingen, Switzerland) that has been less studied. There are many good comparative points discussed within and the overall analyses are fine. I have no important disagreements with the general conclusions reached in this article.

Experimental design

Seems fine. Good figures.

Validity of the findings

The most important issue that this article needs to address is priority. This is not the first time that these fossil tarsals have been described. The authors are unaware that that these tarsal elements, attributed to Caenopithecus, were originally reported in Dagosto's 1986 PhD dissertation. See pages 197-198 for her brief comments, Table 19 (p.204) lists them and Table 22 has several measures for these elements. They are also figured in Fig. 54 (p.227-228). Her functional comments about quadrupedal slow climbing in Caeonopithecus proceed those presented here as well (p. 333). As such, this is not the first time that the first known postcranial elements are being reported for Caenopithecus. It is actually a reanalysis. This aspect of the manuscript needs to be completely rewritten and revised throughout. This manuscript is much more thorough than Dagosto's work in 1986, given the 30 years of work in between, and the authors are welcome to disagree with Dagosto in terms of phyletic closeness with Afradapis, a taxon unknown in 1986, but the original work belongs to her.

I would note two additional cautionary points. If you believe hoanghoniines are sivaladapids, as most would, then the postcranials attributed to Kyitchauaugia are not the first postcranial elements attributed to a sivaladapid (a first metatarsal of Hoanghonius is known since 1999) - a small modification in a single sentence (screen 56).

I would also caution about the hindlimb suspension comments. It is quite common for living primates to hindlimb suspend (this is what makes apes odd via arm suspension). There are many different morphologies associated with these taxa (consider a taxon like Alouatta versus lemurs and galagos). I do not disagree that a taxon like Caeonopithecus could probably use this behavior but the anatomical connections are difficult to isolate (as in most postures).
--DLG

Additional comments

None

·

Basic reporting

- perfect English;
- Introduction and background well presented;
- Relevant literature employed (but some [at least two] are not published and again in review);
- Figures abundant and relevant to the content of the article;
- MS well organized;
- Results relevant to the hypothesis.

Experimental design

- Ms with original primary research within the Scope of the Journal (Biological Sciences);
- Research question clearly defined, relevant and meaningful;
- The investigations were conducted rigorously and to a high technical standard;
- Numerous methods well explained, and relying on previous works;
- The results can be reproducible by other investigators;
- No problem of ethics.

Validity of the findings

- The data are housed in the Naturhistorisches Museum in Basel;
- The conclusions rely on the results and are clearly written. They are in total connection with the original question investigated;
There are some speculations, but they remain relevant (consistent hypothesis) relying on the results obtained;
- Results are new, and perfectly discussed.

Additional comments

In this project of paper for “PeerJ”, Erik Seiffert, Loïc Coster and Doug Boyer report additional postcranial evidence of a medium sized primate from the well-known Egerkingen fissure filling in Switzerland. The fossil material (tarsal) was found in the last century and is housed in the Naturhistorisches Museum in Basel. These tarsal bones (one talus and three calcanei) were found in a locality in which several primates co-occur, notably two medium-sized primates (Leptadapis and Caenopithecus). The challenge was therefore to determine to which taxon these postcranial remains belonged to, inasmuch as they were not find in articulation with other primate postcranial and cranio-dental remains. In this project of paper, the authors perform a bunch of analyses for determining the most likely taxonomic option for these isolated tarsal bones. Their conclusions are rather in favour to Caenopithecus, an option that I follow.

The manuscript is well-organized and particularly well-written (as usual for the authors). Just after a brief introduction, the authors propose an historical and taxonomic background regarding the primates in question, a section which is necessary and interesting (it completes the short introduction). This “History of study” section is followed by a long “Material and method” section, which is detailed but useful to understand the different approaches the authors plan to use for obtaining reliable results. The results are well-presented and includes appropriate/informative figures. I would like to acknowledge the authors for the high quality of the fossil specimen illustrations (CTscans). The figures are very abundant (19), but necessary. The anatomical descriptions and functional interpretations are detailed, strong and useful. I have no disagreement with the interpretation, functional assessments, and general conclusions. I am however somewhat frustrated (but actually not surprised) by the inconsistent phylogenetic results deriving from different reconstruction approaches and assumptions on the characters. But the authors provide interesting discussions and argumentations regarding these problems and underscore the possibilities of convergences.

I consider this MS to be very close of being publishable. The manuscript is somewhat long, but the length is justified for the topics covered with no pruning necessary, and it should be published after addressing the following points:

- the relevant references are cited, but I wonder if the journal accepts to include in the “bibliographic” section, papers that are not published (even accepted) but only “in review”: e.g., Marigo et al., in review to JHE; Yapuncich et al. to ?? . This latter is mentioned as “in review” in the main text (line 146) and in press in the bibliographic section (line 1318), without indication of journal.

- Line 541, regarding sivaladapid postcranial material: the authors mention that “… if correct, the tarsal specimens would be the only known postcranial elements from that clade”. Other postcranial materials of sivaladapids have been mentioned and published: see Marivaux et al., 2008a-b – AJPA and JHE (femur, pelvis, humerus, and calcaneus).

- Line 811, “… instead the lateral tibial facet is mediolaterally broad and…”: lateral fibular facet, no?



There is no doubt that this paper will be of interest to the readers of “PeerJ”. For all these reasons, I highly recommend the publication of this article after minor, somewhat cosmetic revisions!

Laurent Marivaux

Reviewer 3 ·

Basic reporting

This is an incredibly thorough (101 pp on 4 bones) description and quantitative analysis of Caenopithecus lemuroides tarsal morphology, and the topic is certainly appropriate for PeerJ. It is a well-written and well-illustrated manuscript, but it still requires some revisions.

Experimental design

First, and most importantly, an appendix with all of the specimens (extant and extinct) that were examined for this study must be added. Inclusion of a Specimens Examined appendix is necessary for morphological studies so that analyses can be replicated. A complete Specimens Examined appendix will resolve some of the issues in this manuscript (see below).

Line(s):

134: there are some taxon means listed in Appendix S1. Which specimens contributed to those means?

200-202: specimens are abraded so “not all measurements in the database could be taken,” yet some “were estimated due to damage.” Why were some estimated if the specimens were damaged? Doesn’t this defeat the purpose of taking precise measurements? Also, how, exactly, were measurements estimated?

203: Table 2 is cited but I did not receive this table, nor could I download it. What does it include?

207: 18 linear measurements and 6 angular measurements were mentioned above, but only 23 are shown in Appendix S2. Also, they are just numbered 1-23. Without descriptions for these 23 measurements, it is impossible to evaluate them. Such descriptions must be added to Appendix S2 to make it a useful supplement to this manuscript.

210-211: 52 extant individuals and 7 fossil genera are mentioned, but it’s not clear which specimens are included in this analysis because a Specimens Examined list/appendix was not provided (see above). Again, such a list must be added to make this analysis repeatable.

Validity of the findings

No comments.

Additional comments

Line(s):

110 and 117: USNM (117), not NMNH (110), is the correct abbreviation for the United States National Museum of Natural History.

635, 737, 740, 747, 769, 786, 795, 803, 815, 819, 830, 873, 891, 894, 902, 903 (twice): “hindlimb” should be two words, so it should be changed to “hind limb” throughout. The authors do correctly have this as two words in lines 1012-1014, but it must be corrected in the other 17 places it’s used in the manuscript.

796: quotes are used around “tarsal tunnel” but no reference is provided. Who calls this a “tarsal tunnel?” A reference should be cited for this terminology.

1090-1098: why aren’t curators and collection managers who provided access to comparative material acknowledged? Such a detailed quantitative analysis of numerous specimens would not have been possible without access to the museum collections listed in the Institutional Abbreviations section. Just as museum specimens should be listed in a Specimens Examined appendix, those who gave the authors access to these specimens should be listed and thanked in the Acknowledgements section.

---

## Round 0.2 · accepted · Accept

Thank you for your thorough and considerate revision based on the comments of reviewers. The paper is in excellent condition, and represents a very fine, detailed piece of new science.

I'm putting in here several minor comments that the authors and PeerJ production staff should be sure are accounted for during the proofing stage:

Ln. 191, spell out ‘ninety five’ rather than starting it numerically.
Ln. 205: fix syntax – insert ‘be’?
The new stereo pair figures are great; they seem to be of casts. I didn’t see this mentioned, could you be sure that if they are casts, it’s referenced in the figure captions please?

We're very appreciative that you've chosen PeerJ for publication of your study, it's a excellent piece of work.